# The complete structure of the human TFIIH core complex

Basil J Greber[1,2], Daniel B Toso[1], Jie Fang[3], Eva Nogales[1,2,3,4]*

[1]California Institute for Quantitative Biosciences, University of California, Berkeley, United States; [2]Molecular Biophysics and Integrative Bio-Imaging Division, Lawrence Berkeley National Laboratory, Berkeley, United States; [3]Howard Hughes Medical Institute, University of California, Berkeley, United States; [4]Department of Molecular and Cell Biology, University of California, Berkeley, United States

**Abstract** Transcription factor IIH (TFIIH) is a heterodecameric protein complex critical for transcription initiation by RNA polymerase II and nucleotide excision DNA repair. The TFIIH core complex is sufficient for its repair functions and harbors the XPB and XPD DNA-dependent ATPase/helicase subunits, which are affected by human disease mutations. Transcription initiation additionally requires the CdK activating kinase subcomplex. Previous structural work has provided only partial insight into the architecture of TFIIH and its interactions within transcription pre-initiation complexes. Here, we present the complete structure of the human TFIIH core complex, determined by phase-plate cryo-electron microscopy at 3.7 Å resolution. The structure uncovers the molecular basis of TFIIH assembly, revealing how the recruitment of XPB by p52 depends on a pseudo-symmetric dimer of homologous domains in these two proteins. The structure also suggests a function for p62 in the regulation of XPD, and allows the mapping of previously unresolved human disease mutations.
DOI: https://doi.org/10.7554/eLife.44771.001

*For correspondence:
enogales@lbl.gov

Competing interests: The authors declare that no competing interests exist.

## Introduction

Transcription factor IIH (TFIIH) is a 10-subunit protein complex with a total molecular weight of 0.5 MDa that serves a dual role as a general transcription factor for transcription initiation by eukaryotic RNA polymerase II (Pol II), and as a DNA helicase complex in nucleotide excision DNA repair (NER) (*Compe and Egly, 2016*; *Sainsbury et al., 2015*). Mutations in TFIIH subunits that cause the inherited autosomal recessive disorders xeroderma pigmentosum (XP), trichothiodystrophy (TTD), and Cockayne syndrome (CS) are characterized by high incidence of cancer or premature ageing (*Cleaver et al., 1999*; *Rapin, 2013*). Furthermore, TFIIH is a possible target for anti-cancer compounds (*Berico and Coin, 2018*) and therefore of great importance for human health and disease.

The TFIIH core complex is composed of the seven subunits XPB, XPD, p62, p52, p44, p34, and p8, and is the form of TFIIH active in DNA repair (*Svejstrup et al., 1995*), where TFIIH serves as a DNA damage verification factor (*Li et al., 2015*; *Mathieu et al., 2013*) and is responsible for opening a repair bubble around damaged nucleotides. This activity depends on both the SF2-family DNA-dependent ATPase XPB, and the DNA helicase activity of XPD (*Coin et al., 2007*; *Evans et al., 1997*; *Kuper et al., 2014*). TFIIH function in transcription initiation requires the double-stranded DNA translocase activity of XPB to regulate opening of the transcription bubble (*Alekseev et al., 2017*; *Fishburn et al., 2015*; *Grünberg et al., 2012*), and additionally the CdK activating kinase (CAK) complex, which harbors the kinase activity of CDK7 as well as the Cyclin H and MAT1 subunits (*Devault et al., 1995*; *Fisher et al., 1995*; *Fisher and Morgan, 1994*; *Shiekhattar et al., 1995*; *Svejstrup et al., 1995*). Targets of human CDK7 include the C-terminal heptapeptide repeat domain of the largest subunit of Pol II, as well as cell-cycle regulating CDKs (*Fisher and Morgan, 1994*;

**eLife digest** The DNA inside a cell carries the instructions it needs to survive. Living cells use many different proteins to read and maintain this store of information. For example, a group of ten proteins collectively called TFIIH is often involved in both reading and repairing the DNA. Proteins in the TFIIH complex include p52, p62, XPB and XPD.

Understanding the structure of the proteins in TFIIH could reveal much about how it works and how changes to its structure contribute to various medical conditions. Yet TFIIH is a dynamic assembly of molecules and includes many proteins, which makes examining its structure challenging. An ideal protein structure should provide an accurate map of the positions of all the atoms in a protein. Previously, it has not been possible to get this level of detail for TFIIH.

Greber et al. used an approach called cryo-electron microscopy (also called cryo-EM) to reveal the structure of TFIIH collected from human cells. The structure revealed several new details, including how p52 helps XPB attach to the rest of TFIIH, and that p62 helps to control the activity of XPD. With such a detailed structure, Greber et al. could link changes in TFIIH that are seen in different human diseases to specific parts of the complex.

Examining the atomic details of proteins can reveal a lot about how they work and the changes that occur during different diseases. These structures can also help to reveal aspects of how DNA is read and repaired, and may help to design new approaches to treat diseases in the future.
DOI: https://doi.org/10.7554/eLife.44771.002

*Shiekhattar et al., 1995*). MAT1 serves as a bridging subunit that promotes CAK subcomplex formation by interacting with Cyclin H and CDK7 (*Devault et al., 1995*; *Fisher et al., 1995*), recruits the CAK to the core complex by interactions with XPD and XPB (*Abdulrahman et al., 2013*; *Busso et al., 2000*; *Greber et al., 2017*; *Rossignol et al., 1997*), and also aids in Pol II-PIC formation by establishing interactions with the core PIC (*He et al., 2013*; *He et al., 2016*; *Schilbach et al., 2017*). The presence of MAT1 inhibits the helicase activity of XPD (*Abdulrahman et al., 2013*; *Sandrock and Egly, 2001*), but the mechanism of this inhibition is not fully understood. While the enzymatic activity of XPD is not required for transcription initiation, it is critical for the DNA repair function of TFIIH (*Dubaele et al., 2003*; *Evans et al., 1997*; *Kuper et al., 2014*). Therefore, NER requires the release of the CAK subcomplex from the core complex (*Coin et al., 2008*). The activities of both XPB and XPD are regulated by interactions with additional TFIIH components, including that of p44 with XPD (*Coin et al., 1998*; *Dubaele et al., 2003*; *Kim et al., 2015*), and those of the p52-p8 module with XPB (*Coin et al., 2007*; *Coin et al., 2006*; *Jawhari et al., 2002*; *Kainov et al., 2008*). These interactions are likely to be crucial for TFIIH function, as some are affected by disease mutations (*Cleaver et al., 1999*), but they have been only partially characterized mechanistically.

Our previous structure of the TFIIH core-MAT1 complex at 4.4 Å resolution (*Greber et al., 2017*) allowed modeling of TFIIH in the best-resolved parts of the density map, but several functionally important regions remained unassigned or only partially interpreted because reliable de novo tracing of entire domains in the absence of existing structural models was not possible. Here, we present the complete structure of the human TFIIH core complex in association with the CAK subunit MAT1, determined by phase plate cryo-electron microscopy (cryo-EM) at 3.7 Å resolution. Our structure reveals the complete architecture of the TFIIH core complex and provides detailed insight into the interactions that govern its assembly. Additionally, our cryo-EM maps define the molecular contacts that control the regulation of the XPB and XPD subunits of TFIIH, including the critical p52-XBP interaction, and an extensive regulatory network around XPD, formed by XPB, p62, p44, and MAT1.

## Results

### Structure determination of TFIIH

To determine the complete structure of the human TFIIH core complex, we collected several large cryo-EM datasets (*Supplementary file 1*) of TFIIH immuno-purified from HeLa cells using an electron microscope equipped with a Volta phase plate (VPP) (*Danev and Baumeister, 2017*) and a direct electron detector camera mounted behind an energy filter. From a homogeneous subset of

approximately 140,000 TFIIH particle images identified by 3D classification (*Scheres, 2010*), we reconstructed a 3D cryo-EM density map at 3.7 Å resolution (*Figure 1—figure supplements 1* and *2A–C*). This VPP-based cryo-EM map was substantially improved compared to our previous maps obtained without phase plate, both in resolution and interpretability (*Figure 1—figure supplement 2D–G*), and enabled building, refinement, and full validation of an atomic model of the TFIIH core complex and the MAT1 subunit of the CAK subcomplex (*Figure 1A–C*, *Figure 1—figure supplement 2B,C*, *Supplementary file 2*, *3*), while the remainder of the CAK subcomplex is invisible in our map because it is flexibly tethered to the TFIIH core complex. Tracing and sequence register assignment of protein components modeled de novo was facilitated by density maps obtained from

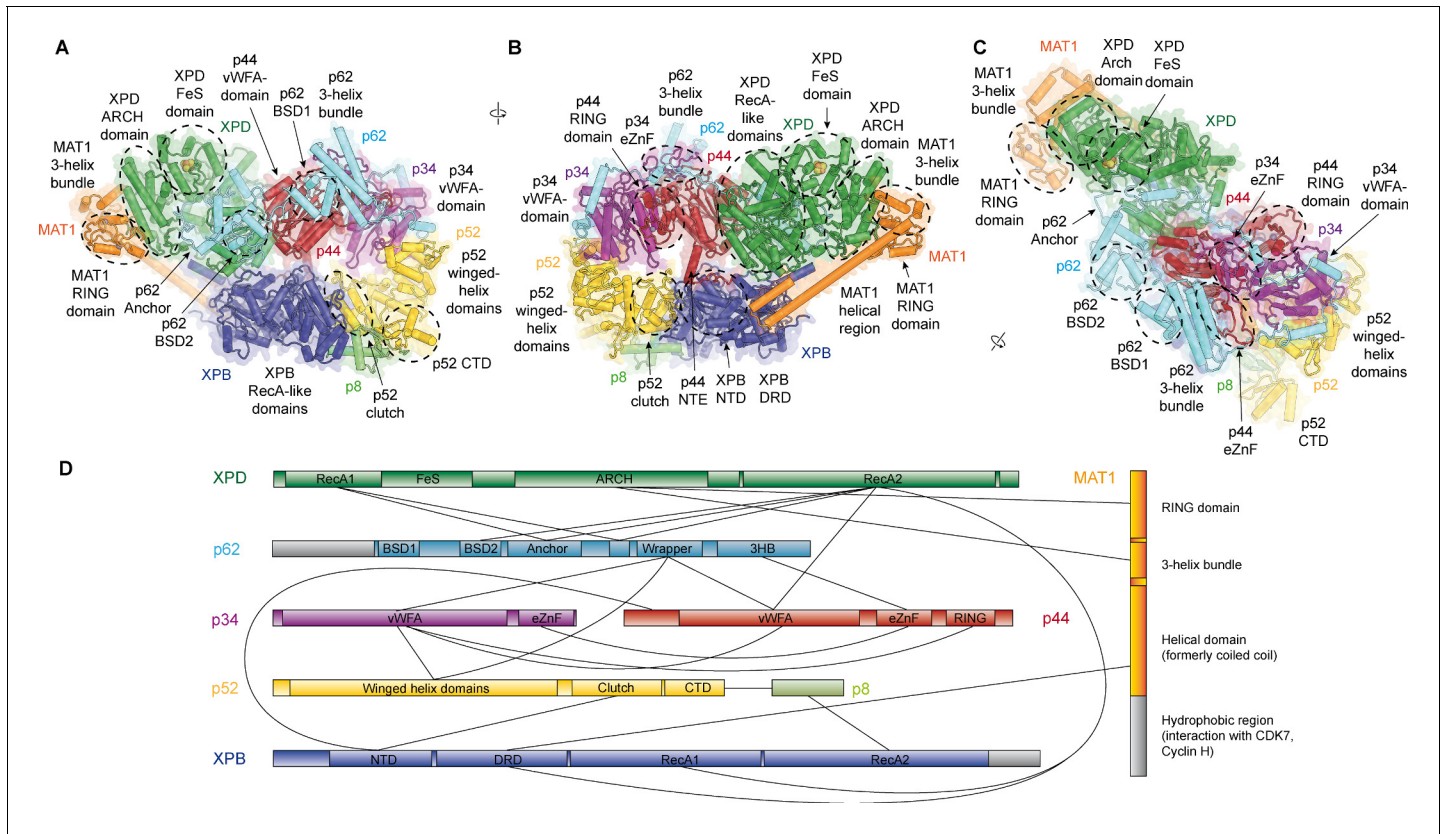

**Figure 1.** Structure of the TFIIH core complex. (A, B, C) Three views of the structure of the TFIIH core complex and MAT1. Subunits are color-coded and labeled (in color); individual domains are labeled (in black) and circled if needed for clarity. (D) Domain-level protein-protein interaction network between the components of the TFIIH core complex and MAT1 derived from the interactions observed in our structure. Proteins are shown with the same colors as in A and major unmodeled regions are shown in grey. Abbreviations: CTD: C-terminal domain; DRD: DNA damage recognition domain; FeS: iron sulfur cluster domain; NTD: N-terminal domain; vWFA: von Willebrand Factor A.

DOI: https://doi.org/10.7554/eLife.44771.003

The following figure supplements are available for figure 1:

**Figure supplement 1.** Sample micrograph, sample 2D classes, and data processing scheme for global reconstruction.
DOI: https://doi.org/10.7554/eLife.44771.004

**Figure supplement 2.** Resolution estimation, validation statistics, and quality of the density.
DOI: https://doi.org/10.7554/eLife.44771.005

**Figure supplement 3.** Focused classification of the p62 BSD2 domain.
DOI: https://doi.org/10.7554/eLife.44771.006

**Figure supplement 4.** Focused classification and interpretation of the MAT1 RING domain density.
DOI: https://doi.org/10.7554/eLife.44771.007

**Figure supplement 5.** Multibody refinement.
DOI: https://doi.org/10.7554/eLife.44771.008

**Figure supplement 6.** Detailed architecture of the TFIIH core complex.
DOI: https://doi.org/10.7554/eLife.44771.009

focused classification and multibody refinement (*Figure 1—figure supplements 3–5*) (*Bai et al., 2015*; *Nakane et al., 2018*), which resulted in density maps of improved interpretability for all three sub-volumes and a slightly improved resolution of 3.6 Å for the XPD-MAT1 region. Both the overall and multibody-refined maps showed clear side chain information (*Figure 1—figure supplement 6A–D*). Furthermore, our model was corroborated by existing chemical crosslinking-mass spectrometry (CX-MS) data of human TFIIH (*Luo et al., 2015*) and site-specific crosslinks from yeast TFIIH (*Warfield et al., 2016*) (*Figure 1—figure supplement 6E–I*, *Supplementary file 4*).

## Detailed architecture of TFIIH and structure of p62

Our structure of the TFIIH core complex shows its horseshoe-like overall shape (*Figure 1A–C*, *Video 1*), as observed in previous lower-resolution reconstructions of free and PIC-bound TFIIH (*Gibbons et al., 2012*; *Greber et al., 2017*; *He et al., 2016*; *Murakami et al., 2015*; *Schilbach et al., 2017*), and allows us to define the complete set of inter-subunit interactions that lead to the formation of the TFIIH core complex directly from our structure (*Figure 1D*).

The largest subunits of the complex, the SF2-family DNA-dependent ATPases XPB and XPD, both containing two RecA-like domains (RecA1 and RecA2), interact directly (*Greber et al., 2017*), are on one side of the complex, and are additionally bridged by MAT1 (*Figure 1B*), which has been shown to interact with either ATPase in isolation (*Busso et al., 2000*). On the side facing away from MAT1, XPD interacts with the von Willebrand Factor A (vWFA) domain of p44 (*Coin et al., 1998*; *Dubaele et al., 2003*; *He et al., 2016*; *Kim et al., 2015*), which in turn forms a tight interaction with p34 via interlocking eZnF domains (*Schilbach et al., 2017*) and a p44 RING domain interaction (*Radu et al., 2017*) (*Figure 1B–D*, *Figure 1—figure supplement 6J,K*), consistent with the formation of a multivalent interaction network between p34 and p44 (*Radu et al., 2017*). The vWFA domain of p34 recruits p52 by a three-way interaction that involves the most N-terminal winged helix domain in p52 and a helical segment of p62 (*Schilbach et al., 2017*) (*Figure 1—figure supplement 6L*). The p52 C-terminal region comprises two domains; first, the 'clutch' that interacts with XPB (*Jawhari et al., 2002*) and second, a dimerization module that binds p8 (*Kainov et al., 2008*), thereby recruiting XPB to TFIIH and cradling XPB RecA2 (see below). In addition to this structural framework that is formed by folded domains, our cryo-EM map reveals several interactions involving extended protein segments, including several interactions formed by p62 (*Figure 2*), and an interaction between the p44 N-terminal extension (NTE) and the N-terminal domain (NTD) of XPB (*Figure 1B*). To form this interaction, approx. 15 residues of p44 span the distance between the p44 vWFA domain and the XPB NTD, where a small helical motif in p44 contacts XPB residues 72–75, 95–102, and 139–143, in agreement with CX-MS data (*Luo et al., 2015*) (*Figure 1—figure supplement 6E*). Partial deletion of the p44 NTE in yeast causes a slow-growth phenotype, suggesting a functional role for this p44-XPB interaction (*Warfield et al., 2016*).

The p62 subunit is almost completely resolved in our structure and exhibits a complex beads-on-a-string-like topology. It fully encircles the top surface of TFIIH (*Figures 1C* and *2A*, *Figure 2—figure supplement 1*), interacting with XPD, p52, p44, and p34, in agreement with previous structural findings (*Greber et al., 2017*; *Schilbach et al., 2017*). Based on these interactions, p62 can be subdivided into three functional regions: (i) the N-terminal PH-domain, disordered in our structure, is responsible for mediating interactions with components of the core transcriptional machinery (*Di Lello et al., 2008*; *He et al., 2016*; *Schilbach et al., 2017*), transcriptional regulators (*Di Lello et al., 2006*),

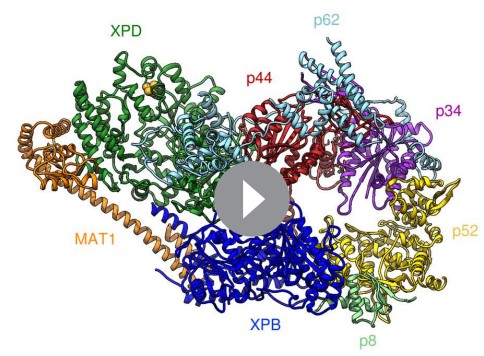

Overall architecture of TFIIH

**Video 1.** Architecture of the TFIIH core complex. Rotating structure of the TFIIH core complex, followed by views that highlight the interactions of p62 near the nucleotide binding pocket of XPD and near the substrate binding cleft of XPD (binding sites are indicated by a flashing ADP molecule and DNA strand, respectively). Bound substrates, which are not present in our structure, were superposed from PDB ID 6FWS (*Cheng and Wigley, 2018*).
DOI: https://doi.org/10.7554/eLife.44771.010

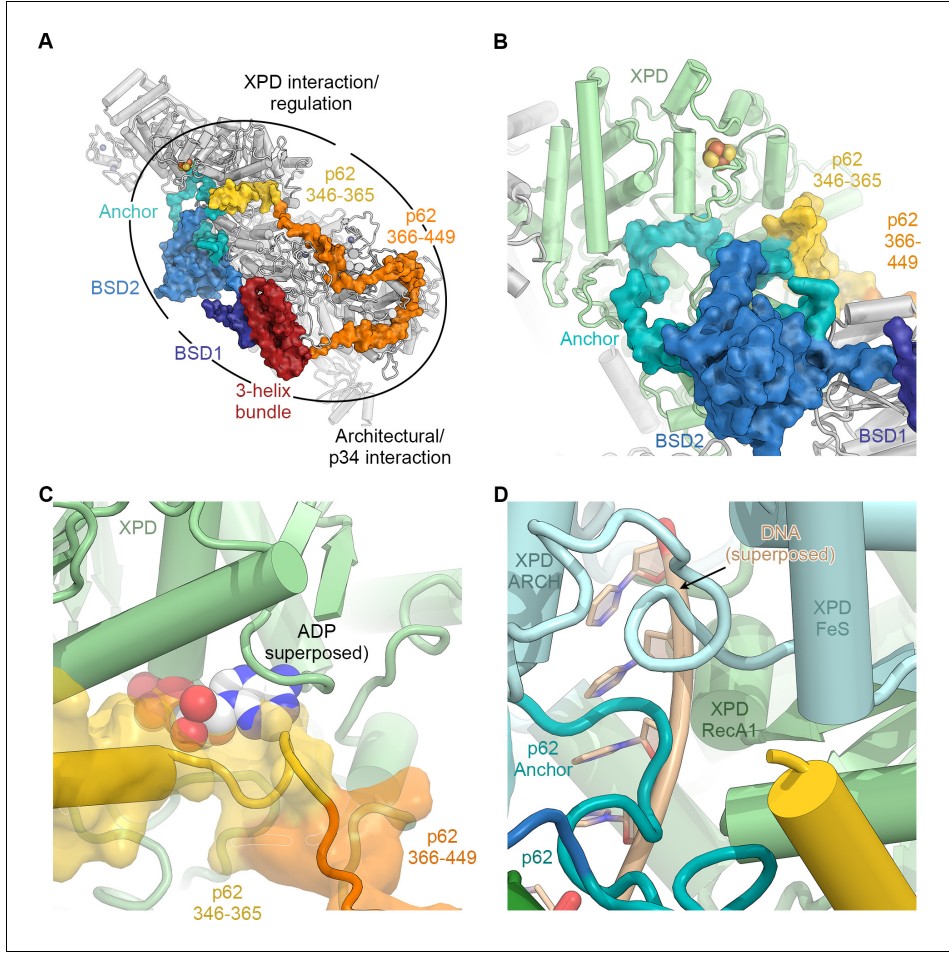

**Figure 2.** The structure of p62. (**A**) View of the top surface of the TFIIH core complex; p62 is color-coded by structural region. (**B**) The BSD2 (blue) and XPD anchor segments (teal) of p62 (surface) interact with the region around the XPD substrate-binding cavity. (**C**) Residues 346–365 of p62 (yellow) approach the nucleotide-binding pocket of XPD. ADP superposed from the structure of the DinG helicase (*Cheng and Wigley, 2018*). (**D**) Superposition of DNA from the structure of the substrate-bound DinG helicase (*Cheng and Wigley, 2018*) shows that the positions of p62 and RecA1-bound DNA overlap.

DOI: https://doi.org/10.7554/eLife.44771.011

The following figure supplement is available for figure 2:

**Figure supplement 1.** Analysis of the structure of p62.
DOI: https://doi.org/10.7554/eLife.44771.012

and DNA repair pathways (*Gervais et al., 2004*; *Lafrance-Vanasse et al., 2013*; *Okuda et al., 2017*); (ii) residues 108–148 and 454–548 of p62, including the first BSD (BTF2-like, synapse-associated, DOS2-like) domain (BSD1) and the C-terminal 3-helix bundle, play an architectural role by binding to p34 and the extended zinc finger (eZnF) domain of p44 (*Figure 2A*, *Figure 1—figure supplement 6J–L*, *Figure 2—figure supplement 1B,C*); and (iii) residues 160–365, including the BSD2 domain, are responsible for interactions with and regulation of XPD (*Figure 2B–D*, *Figure 2—figure supplement 1D–F*).

Specifically, p62 residues 160–365 form three structural elements that interact with XPD (*Figure 2B*, *Video 1*), in agreement with previous biochemical, structural, and CX-MS data (*Figure 1—figure supplement 6G*) (*Jawhari et al., 2004*; *Luo et al., 2015*; *Schilbach et al., 2017*). First, an α-helix formed by p62 residues 295–318 binds directly to XPD RecA2 and thereby recruits residues 160–258 of p62, comprising the BSD2 domain and adjacent sequence elements, to this surface of XPD RecA2 (*Figure 2A,B*, *Figure 2—figure supplement 1D*). Second, p62 residues 266–287 are inserted into the DNA-binding cavity of XPD (*Figure 2B,D*), in agreement with previous observations

(*Schilbach et al., 2017*). This inserted p62 segment directly blocks a DNA-binding site on XPD RecA1 (*Figure 2D*) and localizes near the access path to a pore-like structure between the XPD FeS and ARCH domains. While p62 does not directly contact the DNA-binding surface on XPD RecA2, it may still sterically interfere with DNA binding or access to the helicase elements of XPD in this region (*Figure 2—figure supplement 1E*). Therefore, this segment of p62 may need to move away when XPD binds and unwinds DNA. Third, p62 residues 350–358 form a short α-helix that binds in a cleft between the two RecA-like domains of XPD (*Figure 2C*), so that it not only closes the entrance to the nucleotide binding pocket in XPD RecA1 (*Figure 2—figure supplement 1F*), but also partially overlaps with the predicted location of the nucleotide itself (*Figure 2C*), strongly suggesting a role for this p62 sequence element in XPD regulation. The density for these structural elements of p62 (residues 260–300 and 346–365) in our cryo-EM map is weaker than for the remainder of the complex, suggesting a dynamic interaction with XPD that enables them to modulate the access to the nucleotide-binding pocket, the DNA-binding cavity, and the DNA-translocating pore of XPD, depending on the functional state of TFIIH. 3D reconstructions of TFIIH classified for these regions of p62 (*Figure 1—figure supplement 3*) show globally intact TFIIH, both in the presence and absence of the p62 segments at these XPD sites (*Figure 2—figure supplement 1G–J*), supporting our hypothesis of dynamic regulation, rather than the alternative hypothesis of p62 binding to XPD as a requirement for TFIIH stability (*Luo et al., 2015*).

## Molecular basis of XPB recruitment by p52

Our structure of TFIIH resolves the structure and interactions of all four folded domains of human XPB – two RecA-like domains that form the SF2-family type helicase cassette, a DNA damage recognition domain (DRD)-like domain, and an N-terminal domain (NTD) (*Figure 3A*, *Figure 3—figure supplement 1*) – and reveals the molecular basis of XPB recruitment by p52. The XPB NTD encompasses residues 1–165, with the first 55 residues forming an N-terminal extension (NTE), and the remainder assuming a mixed α/β-fold with four α-helices and five β-strands (*Figure 3A*). The side chain densities in the cryo-EM map (*Figure 1—figure supplement 6A*) and CX-MS data (*Luo et al., 2015*) (*Figure 1—figure supplement 6F*) both confirm our assignment of this domain. Existing biochemical data show that the XPB NTD is required for integration of XPB into TFIIH (*Jawhari et al., 2002*) by forming an interaction with p52 that has been referred to as the 'clutch' (*Schilbach et al., 2017*). In our structure, the p52 contribution to the clutch encompasses p52 residues 306–399, which, strikingly, assume the same overall fold as the XPD NTD (*Figure 3B*), as hypothesized previously (*He et al., 2016*; *Luo et al., 2015*), thereby forming a pseudo-symmetric dimer of structurally homologous domains. The two domains interact through their β-sheets, via both hydrophobic and charged interactions (*Figure 3—figure supplement 2A–C*), and with the most N-terminal β-strand emanating from the XPB NTD extending the p52 β-sheet by additional lateral interactions (*Figure 3A,C*).

Our structural findings rationalize biochemical data that show that deletion of XPB residues 1–207, but not deletion of residues 1–44, impairs the p52-XPB interaction (*Jawhari et al., 2002*) (*Figure 3C*). Our structure is also consistent with data indicating that p52 residues 304–381 are critical for the XPB-p52 interaction (*Coin et al., 2007*; *Jawhari et al., 2002*), but does not show any contacts that could explain that reported binding of XPB to p52 residues 1–135 or 1–304 (*Jawhari et al., 2002*) (*Figure 3—figure supplement 2D,E*). The interaction between p52 and XPB not only recruits XPB to TFIIH, but also stimulates its ATPase activity in vitro (*Coin et al., 2007*). Because our structure does not shown any elements of p52 approaching the XPB nucleotide-binding pocket, we propose that this effect is likely induced by the interactions of p52 with the XPB NTD and RecA2, which may, together with p8 (*Coin et al., 2006*), properly arrange the XPB helicase cassette to bind and hydrolyze ATP (*Figure 3D*) (*Grünberg et al., 2012*).

The XPB NTD is the site of the two human disease mutations F99S and T119P, which cause XP and TTD, respectively (*Cleaver et al., 1999*). Or structure shows that neither of these residues is in direct contact with p52 or the RecA-like domains of XPB, suggesting that the F99S and T119P mutations exert their detrimental effects through structural perturbation of the XPB NTD (*Figure 3—figure supplements 1B* and *2F–I*). Specifically, T119 is located near a turn at the end of a β-strand (*Figure 3—figure supplement 2G*), where its side chain points towards the solvent. Nevertheless, this residue is highly conserved in eukaryotic XPB from ciliates to humans, in some archaeal and bacterial XPB homologs (*Figure 3—figure supplement 1B*), and in the structurally homologous clutch

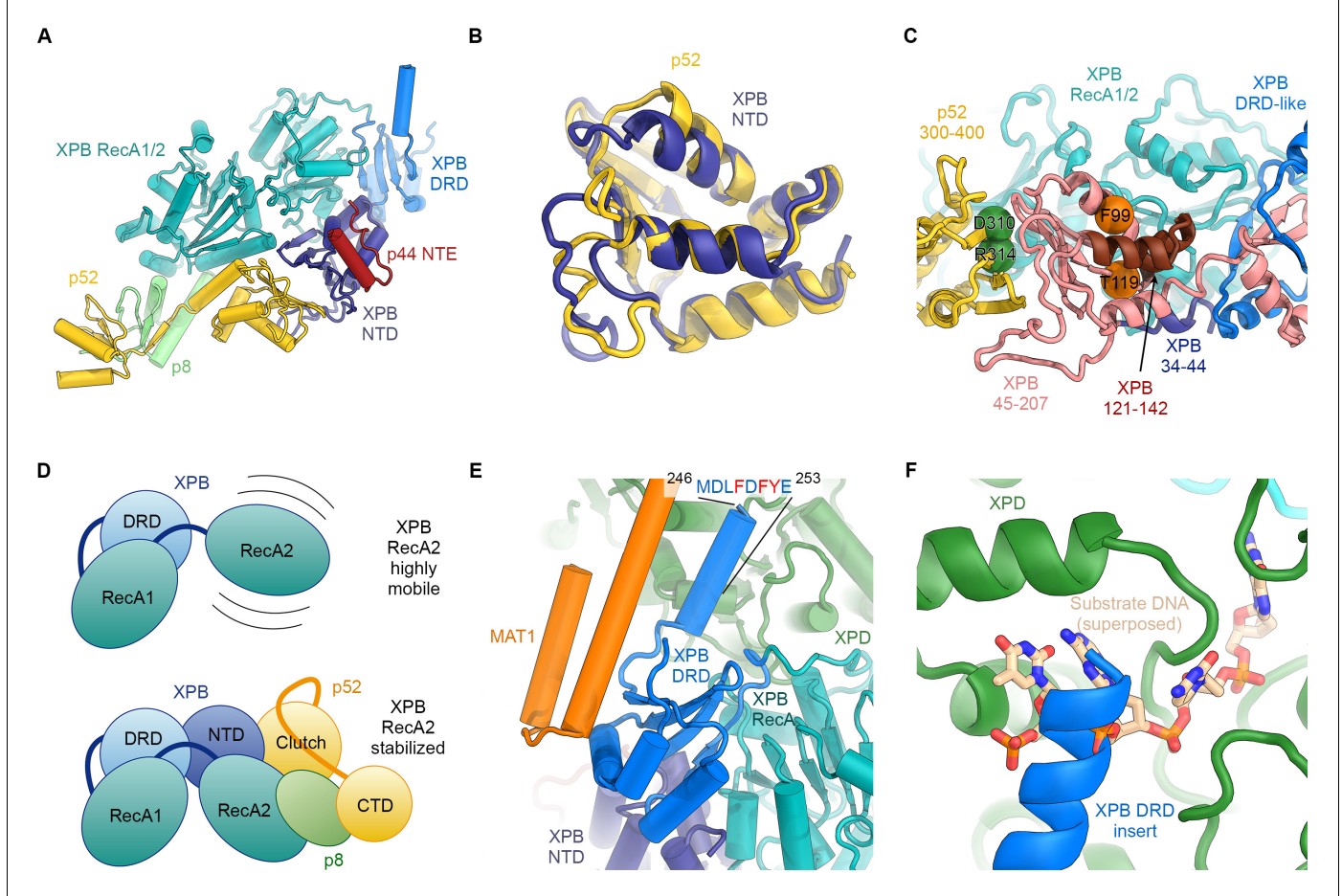

**Figure 3.** Structure and interactions of XPB. (**A**) Bottom lobe of TFIIH. XPB RecA1/2 teal, DRD blue, NTD dark blue, p52 yellow, p8 green, p44 NTE red. (**B**) Superposition of the XPB NTD and the p52 clutch domain. (**C**) Mapping of mutations on the XPB NTD and the p52 clutch domain; mutated regions are color-coded or shown as spheres (see text for details). (**D**) The combined interactions of the p52 clutch, the p8-p52 CTD dimer, and the XPB NTD with XPB RecA2 may restrict the conformational flexibility of XPB RecA2 to optimize XPB activity. (**E**) An extension of the DRD (blue) contacts XPD (green). The sequence for which formation of an α-helix is predicted (*Kelley et al., 2015*) is indicated. (**F**) The DRD extension overlaps with the substrate-binding site on XPD RecA2. Substrate DNA modeled from PDB ID 5HW8 (*Constantinescu-Aruxandei et al., 2016*).

DOI: https://doi.org/10.7554/eLife.44771.013

The following figure supplements are available for figure 3:

**Figure supplement 1.** Domain organization of XPB and sequence alignment of the N-terminal regions of XPB or XPB-like enzymes from eukaryotes, archaea, and bacteria.

DOI: https://doi.org/10.7554/eLife.44771.014

**Figure supplement 2.** Structure and interactions of the XPB DRD and NTD.

DOI: https://doi.org/10.7554/eLife.44771.015

domain in p52 (*Figure 3—figure supplement 2F–H*). This conservation suggests that a threonine at this position is important for the efficient folding of this domain in general, and that the T119P mutation may cause its destabilization, resulting in lower levels of active enzyme in TTD patients. Lower overall levels of properly assembled TFIIH have been shown to be a hallmark of TTD (*Botta et al., 2002*; *Dubaele et al., 2003*; *Giglia-Mari et al., 2004*) and could explain the disease-causing effect of T119P in vivo even though recombinant TFIIH carrying this mutation retains some activity in both transcription initiation and NER (*Coin et al., 2007*). A less likely alternative, given the conservation of the equivalent residue in the p52 clutch, is that T119 is involved in an interaction with a factor that is critical for cellular function, for example in NER.

The F99S mutation affects a residue that is conserved throughout eukaryotic XPB (*Figure 3—figure supplement 1B*), is inserted into a conserved hydrophobic pocket, and localizes to an α-helix at

the XPB contact site with the p44 N-terminal extension (*Figure 3—figure supplement 2I*). This mutation is likely to impair the stability and folding of the XPB NTD. Unlike T119P, this mutation leads to impaired DNA opening in NER assays, reduced interaction with p52, reduced ATPase activity (*Coin et al., 2007*), and strong impairment in DNA damage repair (*Riou et al., 1999*), suggesting a severe effect on the structure of the XPB NTD.

Natural and synthetic mutations in the *Drosophila melanogaster* homolog of p52 that lead to disease-like phenotypes in flies and have similar defects when introduced into human cells (*Fregoso et al., 2007*) map directly to the p52-XPB interface, explaining their detrimental phenotypes (*Figure 3C*, *Figure 3—figure supplement 2A*).

Our structure assigns XPB residues 165–300 to a DRD-like domain that connects the NTD to the RecA-like domain (*Figure 3A*, *Figure 3—figure supplement 1A*), the deletion of which is lethal in yeast (*Warfield et al., 2016*). The DRD is a DNA-binding domain found in DNA repair enzymes and chromatin remodelers (*Mason et al., 2014*; *Obmolova et al., 2000*) and has been implicated in DNA damage recognition in archaeal XPB (*Fan et al., 2006*; *Rouillon and White, 2010*). Our 3.7 Å-resolution map of TFIIH reveals that in eukaryotic XPB, one β-strand of the DRD of archaeal XPB is replaced by an insertion of approximately 70 residues that exhibits relatively low sequence conservation (*Figure 3E*, *Figure 3—figure supplement 1B*) and shifts the domain boundaries of the human XPB DRD-like domain with respect to previous sequence alignments (*Fan et al., 2006*; *Oksenych et al., 2009*). The part of this insertion resolved in our map consists of a negatively charged linker and an α-helical element that contacts XPD directly (*Figure 3E,F*). The surface on XPD involved in this interaction has been implicated in the initial step of DNA substrate binding by XPD (*Constantinescu-Aruxandei et al., 2016*; *Kuper et al., 2012*). Density features and secondary structure prediction indicate the presence of several aromatic side chains of XPB near the interface (*Figure 3E*), where they might form contacts resembling those of nucleoside bases of XPD-bound DNA substrates (*Figure 3F*). Thus, XPB may modulate substrate binding by XPD, further reinforcing the idea that XPD activity is regulated by several other components of TFIIH.

## Conformational dynamics of the TFIIH core complex

In order to investigate the dynamics of TFIIH, we analyzed the conformational landscape of the particles in our cryo-EM dataset (*Figure 4A*, *Figure 4—figure supplement 1*; see Materials and methods for details) (*Nakane et al., 2018*). This analysis revealed the relative motions of the two ATPases and their domains (*Figure 4A*). The major mode of motion, which involves the breaking of the interaction between XPB and XPD, resembles the conformational transition of TFIIH when it enters the Pol II-PIC and binds to DNA (*Greber et al., 2017*; *He et al., 2016*; *Schilbach et al., 2017*) (*Figure 4—figure supplement 2A–C*, *Video 2*). Analysis of our TFIIH structure and comparison with that of the complex within the Pol II-PIC maps (*He et al., 2016*; *Schilbach et al., 2017*) allowed us to identify a specific rearrangement at the interface between the clutch and adjacent winged helix domain in p52 (*Figure 4B*) as the basis of this conformational change. A structural unit composed of XPB, p8 and the clutch domain of p52 undergoes a downward motion upon DNA-binding within the Pol II-PIC (*Figure 4B*, *Figure 4—figure supplement 2B,C*). This conformational change in TFIIH upon PIC entry also appears to break the interaction between MAT1 and the XPB DRD-like domain (*Figure 4C*), which in turn might serve to enable positioning of the CDK7-cyclin H dimer within the CAK subcomplex at the appropriate location for Pol II-CTD phosphorylation in the mediator-bound Pol II-PIC (*Figure 4D*) (*Robinson et al., 2016*; *Schilbach et al., 2017*). Our structural comparison also reveals that a TFIIE-XPB interaction that has been implicated in XPB regulation (*Schilbach et al., 2017*) may depend on the existence of the open conformation of TFIIH, as there would be steric hindrance in a complex involving the closed conformation of TFIIH (*Figure 4—figure supplement 2D–F*).

## Structure of XPD

The structure of XPD shows the conserved domain arrangement of two RecA-like domains (RecA1 and RecA2), with the FeS and ARCH domain insertions in RecA1 (*Constantinescu-Aruxandei et al., 2016*; *Fan et al., 2008*; *Kuper et al., 2012*). The quality of the map allowed us to interpret the density for the N- and C-termini of XPD, which closely approach each other near the nucleotide-binding site within RecA1 (*Figure 5A*). The N-terminus of XPD forms a short two-stranded β-sheet near the

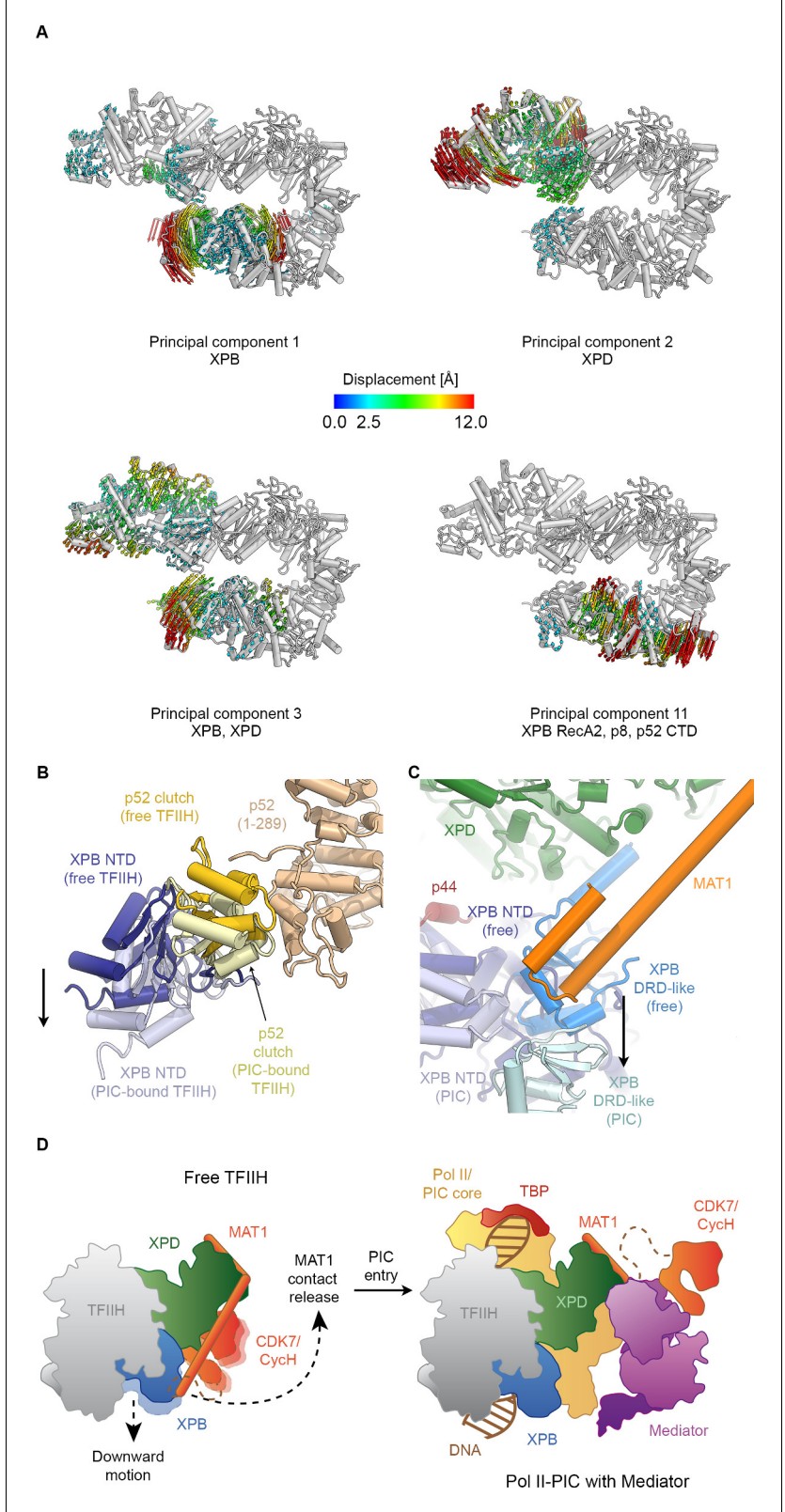

**Figure 4.** Conformational dynamics of TFIIH. (**A**) Results of multibody analysis (also see *Figure 4—figure supplement 1* and Materials and methods for details). Several major modes of motion (Cα displacement indicated by colored arrows; distances < 2.5 Å are not shown) involve the enzymatic subunits of the TFIIH core complex or their domains. (**B**) Motion of the p52 clutch domain (closed conformation gold, open conformation light yellow)
*Figure 4 continued on next page*

*Figure 4 continued*

and associated XPB NTD (closed conformation blue, open conformation light blue) relative to the remainder of p52 (brown), based on comparison of free and PIC-bound TFIIH structures and fitting of domains into Pol II-PIC cryo-EM maps (*He et al., 2016*; *Schilbach et al., 2017*). (**C**) XPB motions from the closed conformation (free TFIIH; darker blue hues) and the open conformation (TFIIH-PIC; lighter hues). The MAT1-XPB contact probably dissociates during this rearrangement. (**D**) Schematic model for the conformational transitions in MAT1 and repositioning of the CAK kinase module during Pol II-PIC entry of TFIIH.

DOI: https://doi.org/10.7554/eLife.44771.016

The following figure supplements are available for figure 4:

**Figure supplement 1.** Analysis of conformational variance of TFIIH.
DOI: https://doi.org/10.7554/eLife.44771.017

**Figure supplement 2.** Conformational dynamics of TFIIH and comparison with the PIC-bound TFIIH.
DOI: https://doi.org/10.7554/eLife.44771.018

nucleotide-binding site in XPD RecA1 (which is empty in our structure) and may contribute to the stabilization of the bound nucleotide via the aromatic side chains of Y14 and Y18, the latter being affected by the Y18H mutation in an XP/TTD patient (*Kralund et al., 2013*) (*Figure 5—figure supplement 1A–C*). The XPD C-terminal segment runs along the side of XPD RecA2 and interacts with the linker between RecA1 and RecA2 (*Figure 5—figure supplement 1D*). The C-terminal segment includes the site of the K751Q polymorphism (*Figure 5—figure supplement 1D*), and deletion of this terminal segment causes XP in human patients (*Cleaver et al., 1999*).

Before XPD-bound DNA reaches the helicase motifs in the RecA like domains, it passes through a pore-like structure next to the 4FeS cluster at the interface between the FeS and ARCH domains (*Figure 5—figure supplement 1E,F*) (*Cheng and Wigley, 2018*; *Constantinescu-Aruxandei et al., 2016*; *Kuper et al., 2012*; *Liu et al., 2008*; *Wolski et al., 2008*). This region was poorly defined in previous TFIIH reconstructions, but our cryo-EM map now shows side-chain densities for the aromatic residues Y158, F161, and F193, which are critical for the DNA-binding, ATPase, and helicase activities of XPD (*Kuper et al., 2014*), as well as for residues Y192 and R196, which form part of a DNA lesion recognition pocket (*Mathieu et al., 2013*) (*Figure 5B*). This functionally important region is only partially conserved in archaeal XPD homologs (*Figure 5—figure supplement 1G–I*) (*Fan et al., 2008*; *Kuper et al., 2012*; *Wolski et al., 2008*). A eukaryotic-specific loop insertion in the XPD ARCH domain (*Greber et al., 2017*; *Schilbach et al., 2017*) closely approaches this binding pocket (*Figure 5B*) and may serve to regulate the binding of DNA in the lesion recognition pocket such as to prevent untimely access of substrates to the XPD pore.

## Interactions and regulation of XPD

Our structure of TFIIH shows that XPD forms architectural and regulatory interactions with four other TFIIH subunits: XPB, p62, p44, and MAT1, which together form a cradle-like structure around XPD (*Figure 5C*). We described above two interactions that could potentially regulate XPD activity: the newly defined interaction of an insertion element in the XPB DRD with a DNA-binding site in XPD (*Figure 3E,F*); and XPD-p62 interactions involving the XPD nucleotide binding pocked and DNA binding cavity (*Figure 2*, *Figure 2—figure supplement 1D–F*)

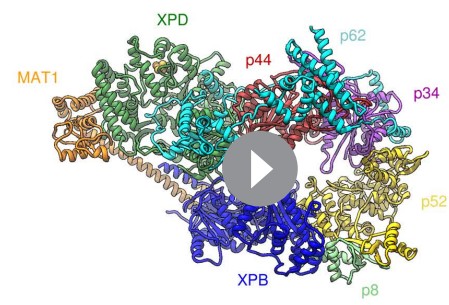

Closed conformation of free TFIIH

**Video 2.** Conformational rearrangements of TFIIH during incorporation into the Pol II-PIC. Transition of TFIIH from the closed conformation observed in our structure of free TFIIH to the open conformation present in Pol II-PIC bound TFIIH, followed by a depiction of TFIIH in the context of the Pol II-PIC. Coordinates (PDB 5IYB, PDB 5OQJ) and maps (EMD-8131, EMD-8132, EMD-3846) used for core-Pol II-PIC depictions and analysis of TFIIH conformation from (*He et al., 2016*; *Schilbach et al., 2017*).
DOI: https://doi.org/10.7554/eLife.44771.019

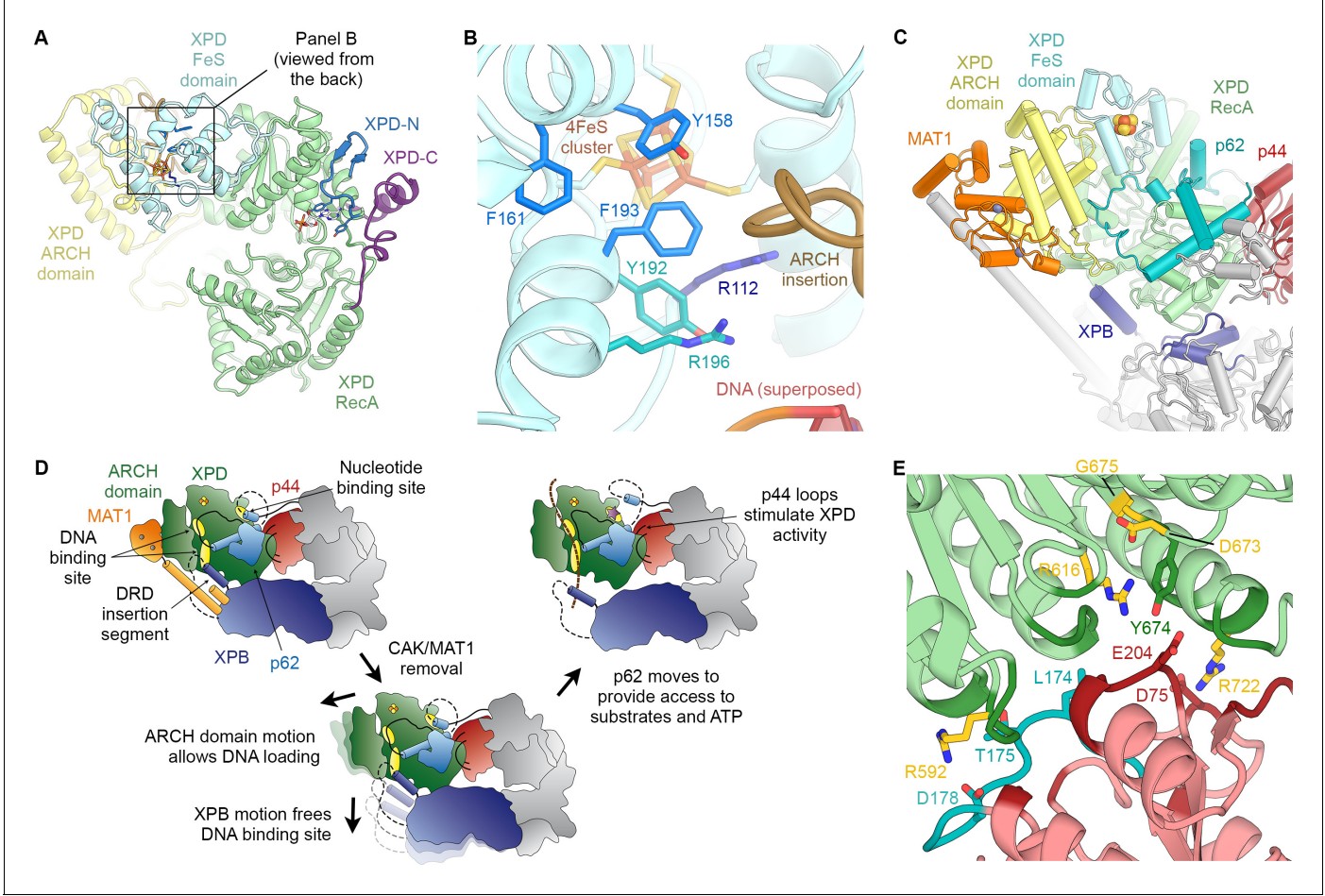

**Figure 5.** Structure and regulation of XPD. (**A**) Structure of XPD colored by domain. N- and C-terminal segments (blue and purple, respectively) of XPD are indicated. An ADP molecule superposed from the structure of DinG (*Cheng and Wigley, 2018*) denotes the nucleotide-binding pocket in XPD RecA1, which is empty in our structure. (**B**) Structure of the FeS domain. Residues critical for XPD enzymatic activity (blue) and damage verification (teal) are indicated. The R112H mutation causes TTD in human patients. ARCH domain insertion brown. DNA superposed from (*Cheng and Wigley, 2018*). The region corresponding to the view in this panel (but viewed from the back side) is indicated in (**A**). (**C**) Interaction network of XPD with surrounding TFIIH subunits (interacting regions colored, remainder grey). (**D**) Cartoon model for repression and de-repression of XPD by MAT1, XPB, and p62. (**E**) XPD-p44 interacting regions (defined as residues within <4 Å of the neighboring protein) are colored in dark green (XPD) and dark red (p44). Residues discussed in the text are shown as sticks; those with mutation data (natural variants or experimental constructs) are colored yellow on XPD, teal on p44. The remainder of the β4-α5 loop harboring the synthetic p44 mutations is colored teal as well.

DOI: https://doi.org/10.7554/eLife.44771.020

The following figure supplements are available for figure 5:

**Figure supplement 1.** Structure of XPD.
DOI: https://doi.org/10.7554/eLife.44771.021

**Figure supplement 2.** XPD-p44 interaction.
DOI: https://doi.org/10.7554/eLife.44771.022

that implicate p62, as well as XPB, in XPD regulation. Additionally, it is known that the helicase activity of XPD is inhibited by the CAK subcomplex (*Araújo et al., 2000*; *Sandrock and Egly, 2001*). The contacts we see between MAT1 and XPD localize to the ARCH domain of XPD and the N-terminal RING domain and helical bundle of MAT1 (residues 1–130) (*Figure 5C*), in agreement with previous structural (*Greber et al., 2017*; *Schilbach et al., 2017*) and biochemical analysis (*Abdulrahman et al., 2013*; *Luo et al., 2015*; *Warfield et al., 2016*). The interaction between the XPD ARCH domain and the MAT1 helical bundle is characterized by charge complementarity (*Figure 5—figure supplement 1J–M*). This interface is highly dynamic, enabling the release of MAT1

and the entire CAK subcomplex from TFIIH during NER, as well as its subsequent re-association to regenerate a transcription-competent TFIIH (*Coin et al., 2008*).

Insertion of substrate DNA into the pore between the XPD ARCH and FeS domains requires the flexibility of the XPD ARCH domain (*Constantinescu-Aruxandei et al., 2016*), and large domain motions have been observed in the structure of the DNA-bound homologous helicase DinG upon nucleotide binding (*Cheng and Wigley, 2018*). This suggests a role for the mobility of the ARCH domain in both DNA loading and DNA translocation by the XPD helicase. Our structure suggests that binding of the MAT1 helical bundle and RING domain to the ARCH domain may prevent such motion and therefore the subsequent substrate loading and XPD helicase activity (*Figure 5D*), in agreement with biochemical data that show XPD inhibition upon MAT1 binding (*Sandrock and Egly, 2001*), as well as reduced single-stranded DNA affinity of TFIIH in the presence of the CAK (*Li et al., 2015*). Conversely, release of MAT1 from XPD might allow the ARCH domain to move more freely, thereby de-repressing XPD. Furthermore, displacement of the MAT1 α-helix that connects XPD to XPB may allow XPB to move away from XPD, thereby unmasking the substrate-binding site on XPD RecA2 that is otherwise occluded by the DRD insertion element (*Figure 5D*). This latter conformational change would be similar, overall, to that seen for TFIIH upon incorporation into the Pol II-PIC, where XPD and XPB move apart and density for the MAT1 helix is missing (*Figure 4C*) (*He et al., 2016*; *Schilbach et al., 2017*). We propose that the combined unmasking of the XPD substrate binding site and the enhanced flexibility of the XPD ARCH domain may both contribute to de-repression of the XPD helicase upon release of MAT1. This mechanism of XPD inhibition by MAT1 does not exclude the possibility of additional repression of NER activity by the CAK subcomplex through phosphorylation of NER pathway components (*Araújo et al., 2000*).

Our structure also resolves in detail the XPD-p44 interaction, a known regulatory interface (*Dubaele et al., 2003*; *Kim et al., 2015*; *Kuper et al., 2014*) affected by numerous disease mutations (*Cleaver et al., 1999*; *Greber et al., 2017*; *Kuper et al., 2014*) (*Figure 5E*). The relatively small interaction surface between p44 and XPD, of just 940 Å$^2$ (*Figure 5—figure supplement 2A*), contrasts with the much larger buried surface of 3300 Å$^2$ between XPD and p62, or 1580 Å$^2$ for the p52-XPB interaction. This smaller interaction surface may result in higher sensitivity to mutations that localize at the XPD-p44 interface. Our structure, thus, rationalizes the deleterious effect of a number of natural and synthetic mutations in this interface (see Appendix 1 and *Figure 5—figure supplement 2B–E*), including mutations L174W and T175R in the β4-α5 loop of p44 (*Figure 5E*) (*Kim et al., 2015*; *Seroz et al., 2000*), which may lead to steric clashes in the densely packed interface (*Figure 5—figure supplement 2B*), and the XPD R722W mutation (*Kuper et al., 2014*), which disrupts the salt bridge with D75 in p44 and may additionally cause steric clashes with neighboring p44 residues due to the bulky tryptophan side chain (*Figure 5—figure supplement 2C*). Our structure also shows that, in contrast to a previously proposed model (*Luo et al., 2015*), the XPD R616P, D673G, and G675R disease mutations act either via disruption of the XPD structure or the XPD-p44 interface, but not via disruption of the interaction with p62 (see Appendix 1 and *Figure 5—figure supplement 2D*). Notably, the p44-dependet stimulation of XPD activity does not depend on the presence of p62 (*Dubaele et al., 2003*; *Kim et al., 2015*; *Kuper et al., 2014*). We were also able to map a number of disease mutations onto our XPD structure (*Figure 6A*, *Video 3*, Appendix 2) and analyze in detail the interactions involving the affected residues (example shown in *Figure 6B*). Our analysis confirms that XP mutations mostly localize near the helicase substrate-binding or active sites, while TTD mutations predominantly localize to the periphery of XPD (*Figure 6*, *Figure 6—figure supplement 1*) (*Fan et al., 2008*; *Liu et al., 2008*), where they disrupt TFIIH assembly and cause the transcription defects that are a hallmark of this disease (*Dubaele et al., 2003*) (Appendix 2).

## Discussion

Our study reveals the complete structure of the TFIIH core complex and provides mechanistic insights into the regulation of its two component helicases. Specifically, it shows XPD wrapped by numerous interactions with XPB, p62, p44, and MAT1 (*Figure 5C,D*), indicating how its activity can be tightly controlled and de-repressed only when its enzymatic function is needed. XPD activity is not needed and most likely inhibited during transcription initiation, but it may also be tightly controlled during NER, when repair bubble opening and lesion verification need to be coordinated with the recruitment and activation of the damage recognition and processing machinery (*Figure 7*).

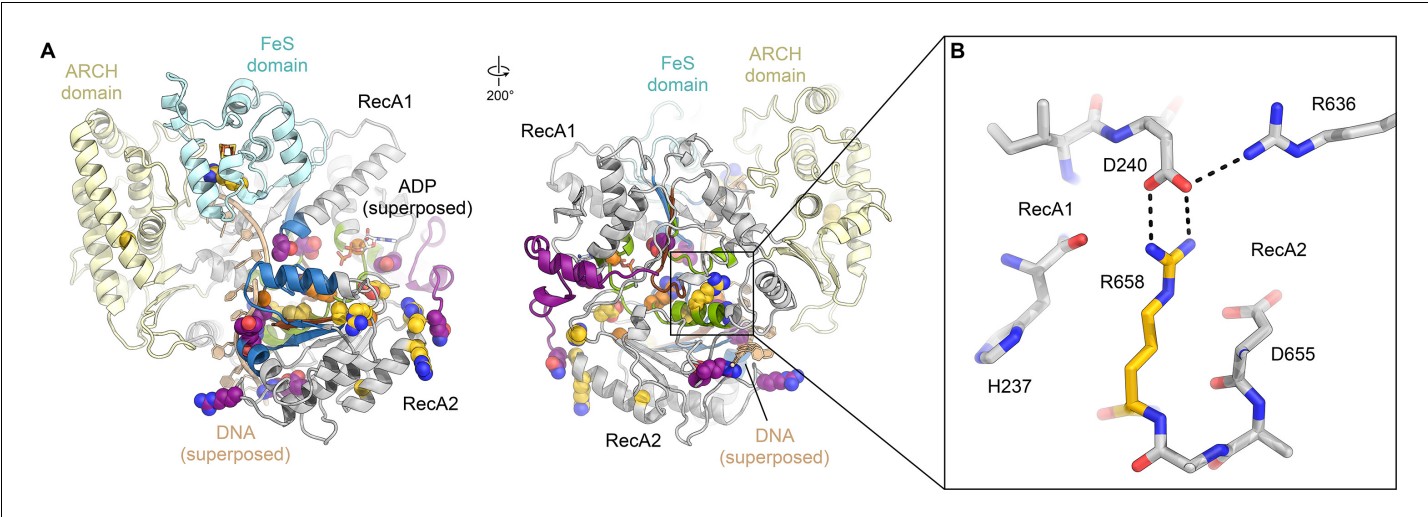

**Figure 6.** Disease mutations in XPD. (A) Residues affected by disease mutations are shown as spheres (XP purple; TTD yellow; CS-XP orange). Conserved helicase elements for DNA binding are shown in blue, for nucleotide binding and hydrolysis in green, and for coupling of nucleotide hydrolysis and DNA translocation in brown. DNA superposed from (*Cheng and Wigley, 2018*). (B) Salt bridge between R658 (RecA2) and D240 (RecA1) visualized in our structure that is affected by the temperature sensitive TTD mutation R658C (*Vermeulen et al., 2001*).

DOI: https://doi.org/10.7554/eLife.44771.023

The following figure supplement is available for figure 6:

**Figure supplement 1.** Mapping of XPD disease mutations on the TFIIH structure.

DOI: https://doi.org/10.7554/eLife.44771.024

While the regulation of XPD by MAT1 and p44 has been studied in some detail, and the domain motions in TFIIH suggest a straightforward mechanism for liberating the substrate-binding site on XPD RecA2, less was known about the interplay between XPD and p62. Our structure now shows how p62 is able to impede both substrate and nucleotide binding in XPD RecA1, and hints at dynamic structural changes of p62 during de-repression and enzymatic activity of XPD, possibly regulated by other components of the transcription or NER pathways.

Our results allow us to put extensive biochemical data on the NER pathway into a structural context (*Figure 7*). Depending on whether XPB binds to the damaged (*Figure 7A*) or undamaged (*Figure 7B*) strand, the combined action of XPD and XPB could lead to the extrusion of a DNA bubble (*Figure 7A*) or to the tracking of the entire complex towards the lesion (*Figure 7B*), which is initially located 3' of the TFIIH binding site (*Sugasawa et al., 2009*). The latter hypothesis is attractive in the context of biochemical data that show that XPD tracks along the damaged strand in the 5' to 3' direction until it encounters the DNA lesion in order to verify the presence of a bona fide NER substrate (*Buechner et al., 2014*; *Li et al., 2015*; *Mathieu et al., 2013*; *Naegeli et al., 1993*;

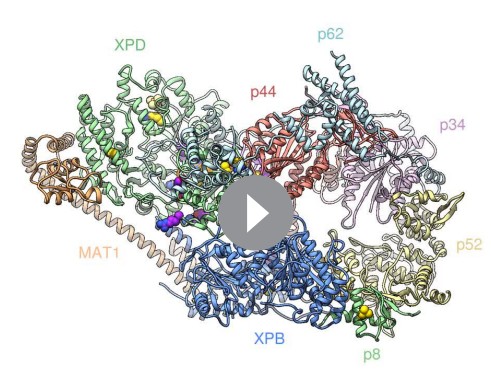

Disease mutations: XP - XP/CS combined - TTD

**Video 3.** Visualization of disease mutations mapped onto the structure of the TFIIH core complex. The human disease mutations discussed in the manuscript are shown in the context of the TFIIH structure. The areas depicted are: (i) The interaction interface between p8 and XPB, affected by a TTD mutation; (ii) the XPB N-terminal domain, affected by XP and TTD mutations; (iii) the active site region of XPD, affected mostly by XP and XP/CS mutations; (iv) the DNA-binding cleft of XPD, affected mostly by XP mutations; (v) the interaction site between XPD and p44, affected mostly by TTD mutations; (vi) the interaction site between MAT1 and XPD, affected by a TTD mutation (see text for further details).

DOI: https://doi.org/10.7554/eLife.44771.025

*Sugasawa et al., 2009*; *Wirth et al., 2016*). It is worth noting that the length of DNA fragments excised during NER is approx. 29 nt, with 22 nt located 5' and 5 nt located 3' of a thymine dimer lesion (*Huang et al., 1992*). According to our structure, the 22 nt 5'-fragment corresponds well to the estimated 20 nt of DNA that are required to span the distance from the DNA damage verification pocket in XPD (*Mathieu et al., 2013*) to the helicase elements of XPB. This proposal is compatible with a model in which TFIIH sitting on the open repair bubble might track towards the lesion, where it would stop due to inhibition of XPD (*Li et al., 2015*; *Mathieu et al., 2013*; *Naegeli et al., 1993*), at which point double incision could be initiated. However, this model (*Figure 7B*) would require strong DNA bending before both XPB and XPD could be loaded. Additionally, it has not been fully resolved whether XPB participates in DNA translocation or unwinding during TFIIH activity in NER (*Li et al., 2015*), which would be required in the tracking model (*Li et al., 2015*), or whether it exclusively acts to anchor the complex in the vicinity of the DNA lesion (*Coin et al., 2007*; *Oksenych et al., 2009*).

Independently of the orientation of the repair bubble, our structural data are compatible with literature data introduced above and a model (*Figure 7C*) that localizes XPG near XPD and p62 (site of 3'-incision), XPF-ERCC1 near XPB (site of 5'-incision), and with RPA binding the non-damaged strand (*Fagbemi et al., 2011*). We have currently not included XPA in this model because its interactions with distinct partners or participation in various processes, such as involvement in CAK release (*Coin et al., 2008*), binding to p8 (*Ziani et al., 2014*), and participation in helicase stalling after lesion recognition (*Li et al., 2015*), suggest its localization to various, often distant sites on TFIIH, or the repair bubble in general (*Sugitani et al., 2016*).

In summary, our structure of the human TFIIH core complex reveals the interactions that govern the architecture and function of this molecular machine, provides new insights into the regulation of its enzymatic subunits, and thus constitutes an excellent framework for further mechanistic studies of TFIIH in the context of larger DNA repair and transcription assemblies.

## Materials and methods

### TFIIH purification, cryo-EM specimen preparation, and data collection

TFIIH was purified and cryo-EM grids were prepared on carbon-coated C-flat CF 4/2 holey carbon grids (Protochips) using a Thermo Fisher Scientific Vitrobot Mk. IV, as previously described (*Greber et al., 2017*). To improve on our previous 4.4 Å cryo-EM map of human TFIIH (*Greber et al., 2017*), which was based on four cryo-EM datasets (3 of which were retained in the 4.4 Å reconstruction, datasets 8–10 in *Supplementary file 1*) from a low-base Titan microscope (Thermo Fisher Scientific) equipped with a side-entry holder (Gatan) and a K2 Summit direct electron detector (Gatan), we collected new data (dataset seven in *Supplementary file 1*) on a Titan KRIOS microscope (Thermo Fisher Scientific) operated at 300 kV extraction voltage and equipped with a $C_S$-corrector, a K2 Summit direct electron detector (Gatan) operated in super-resolution counting mode, and a Quantum energy filter (Gatan). This dataset was collected under the same imaging conditions as our previous data (i.e. 37,879 x magnification resulting in 1.32 Å pixel size, and at a total exposure of 40 e⁻ Å$^{-2}$), except for the change of microscope. Datasets 7–10 could be combined to yield a cryo-EM map at 4.3 Å resolution (not shown), however, this did not lead to a substantial improvement in map quality, suggesting that particle alignment quality was limiting. We therefore opted to collect further data on a Titan KRIOS electron microscope (Thermo Fisher Scientific) operated at 300 kV acceleration voltage and equipped with a Volta Phase Plate (VPP), a Gatan Quantum energy filter (operated at 20 eV slit width), and a Gatan K2 Summit direct electron detector (operated in super-resolution counting mode). VPP data (datasets 1–6 in *Supplementary file 1*) were collected according to the defocus acquisition technique (*Danev and Baumeister, 2017*; *Khoshouei et al., 2017*) at 43,478 x magnification, resulting in a physical pixel size of 1.15 Å on the object scale, with a total electron exposure of 50 e⁻ Å$^{-2}$ at an exposure rate of 6.1 e⁻ Å$^{-2}$ s$^{-1}$ during an exposure time of 8.25 s, dose fractionated into 33 movie frames (50 frames for dataset 6). Data collection was monitored on-the-fly using FOCUS (*Biyani et al., 2017*) to ensure proper evolution of the VPP-induced phase shift.

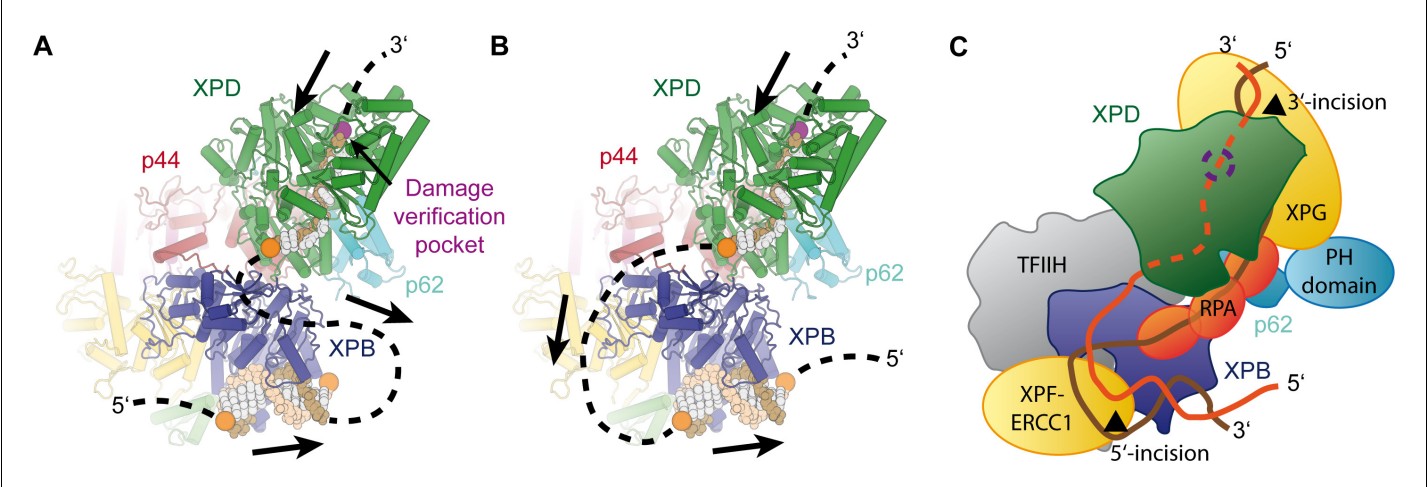

**Figure 7.** Implications for assembly of the repair bubble during NER. (**A**) Schematic of DNA-bound TFIIH (DNA damage verification pocket in XPD and DNA 5′-phosphates indicated by purple and orange spheres, respectively) Binding of both XPB and XPD to the damaged strand would lead to extrusion of a bubble when XPD scans in the 5′−3′ direction, while XPB may be stationary or contribute to bubble extrusion if translocating in the 3′−5′ direction (DNA superposed from PDB IDs 6FWR, 5OQJ (*Cheng and Wigley, 2018*; *Schilbach et al., 2017*)). (**B**) Binding of XPB to the undamaged strand would enable the entire complex to scan in 5′−3′ direction, given the opposing polarities of the two ATPases/helicases involved. (**C**) Model for the assembled repair bubble. Positions of NER factors are approximate. XPG-p62 PH domain interaction according to (*Gervais et al., 2004*). See Discussion for details.

DOI: https://doi.org/10.7554/eLife.44771.026

## Cryo-EM data processing

Initially, we used data collected in 10 microscopy sessions, six sessions using the VPP and four sessions without the VPP, resulting in >30'000 total micrographs, of which approx. 16,000 were retained after inspection of the quality of Thon rings and CTF fitting (for details, see *Supplementary file 1*). Movie stacks were aligned and dose weighed using MOTIONCOR2 (*Zheng et al., 2017*). The aligned, dose weighed sums from the datasets collected at 1.32 Å pixel size (datasets 7–10) were up-sampled to 1.15 Å per pixel to match the scale of the micrographs collected using the VPP (datasets 1–6) after calibrating the two magnifications to each other based on 3D reconstructions computed from the two types of data. CTF parameters were estimated using GCTF (*Zhang, 2016*) and particles were picked using GAUTOMATCH (Kai Zhang, MRC Laboratory of Molecular Biology, Cambridge UK) or RELION (*Scheres, 2015*) using templates generated from a preliminary run without reference templates. All subsequent data processing was performed in RELION 2 (*Kimanius et al., 2016*; *Scheres, 2012*) or RELION 3 (*Nakane et al., 2018*; *Zivanov et al., 2018*).

To remove false positive particle picks and broken particles, an initial 3D classification at low resolution (7.5° angular sampling) was performed on each dataset individually (datasets 3, 4, 5), or on a few pooled datasets if appropriate (datasets 1 and 2 were pooled as they used the same batch of specimen; the non-VPP datasets 7–10 were joined because only few micrographs were retained due to more stringent quality criteria compared to our previous study; and dataset six was initially classified together with particles from dataset four to compensate for particle orientation bias in dataset 6, see *Figure 1—figure supplement 1*). In summary, a total of >2,000,000 initial particle picks were subjected to this initial low-resolution 3D classification, identifying approx. 820,000 intact particles that were subjected to further processing. After 3D auto-refinement and another round of 3D classification, performed separately for the VPP and non-VPP data because the two data types were spuriously separated into distinct classes in combined RELION 3D classification runs, the best classes (one from VPP and non-VPP data each) resulting from the high-resolution 3D classifications were refined according to the gold-standard refinement procedure (fully independent half-sets), resulting in a 3.9 Å-resolution reconstruction according to the FSC = 0.143 criterion (*Rosenthal and Henderson, 2003*; *Scheres and Chen, 2012*). Beam tilt refinement in RELION 3 (*Zivanov et al., 2018*) improved the map computed from the final subset of VPP data (138,659 particle images) to 3.7 Å

resolution. The non-VPP data no longer improved the reconstruction after beam tilt correction and was therefore discarded at this point. The final map was post-processed by application of a B-factor of $-142\ \text{Å}^2$ and low-pass filtration to the nominal 3.7 Å resolution for visualization and later coordinate refinement.

We note that even though the final reconstruction comprises only a relatively small fraction of the total particle picks, the first 3D refinement from 786,755 VPP particle images (*Figure 1—figure supplement 1*) resulted in a 4.3 Å-resolution map that is in excellent agreement with the final map, except for lower resolution and worse map quality caused by residual heterogeneity that was addressed in the subsequent 3D classification step to yield the final set of 138,659 particle images. Therefore, we conclude that our final reconstruction is representative of the overall particle population in the dataset.

To facilitate the interpretation of less ordered or only partially occupied parts of the structure, including the p62 BSD2 domain, the MAT1 RING domain, the MAT1 three-helix bundle at the XPD arch domain, and the N-terminus of XPD, we used signal subtracted classification (*Bai et al., 2015*; *Nguyen et al., 2015*) (p62 BSD2 domain, *Figure 1—figure supplement 3*), focused classification (MAT1 RING domain, *Figure 1—figure supplement 4*), and multibody refinement (*Nakane et al., 2018*) (MAT1 three-helix bundle and XPD N-terminus, *Figure 1—figure supplement 5*). For these classification procedures, we used only the VPP data because 3D classification separated VPP and conventional cryo-EM data into distinct classes, rendering combined classification ineffective. Multibody refinement led to only a slight improvement in resolution for the XPD-MAT1 body (to 3.6 Å) relative to the overall refined best map, and only during the first two iterations, likely due to the relatively small size of the individual bodies and the resulting limited signal for alignment. However, the above-mentioned structural elements showed improved density features (*Figure 1—figure supplement 5B*) and could be more reliably interpreted in the multibody-refined XPD-MAT1 map (green in *Figure 1—figure supplement 5*; also used in *Figure 1—figure supplement 2E,F*). Overall, the use of VPP data in this work resulted in substantial improvements both in nominal resolution (*Figure 1—figure supplement 2D*) and map quality (*Figure 1—figure supplement 2E–G*) compared to our previous 4.4 Å-resolution structure.

## Model building and refinement

The previous structure of the human TFIIH core complex (*Greber et al., 2017*) and of yeast TFIIH in the Pol II-PIC (*Schilbach et al., 2017*) were docked into the cryo-EM map and used as the basis for atomic modeling in O (*Jones et al., 1991*) and COOT (*Emsley et al., 2010*). In addition to these models, the structure of the human p34 VWFA-p44 RING domain complex (*Radu et al., 2017*), the N-terminal RING domain of MAT1 (*Gervais et al., 2001*), the C-terminal RecA-like domain of human XPB (*Hilario et al., 2013*) and several homology models for the p52 winged-helix domains generated using the PHYRE2 web server (*Kelley et al., 2015*) based on templates PDB IDs 3F6O and 1STZ (*Liu et al., 2005*) were used for model building.

### Building of 52

The structure of p52 was traced and built completely de novo, with the exception of the winged-helix domains and the very C-terminal domain, where a homology model was placed into the cryo-EM map together with the p8 structure (*Kainov et al., 2008*; *Vitorino et al., 2007*) and adjusted to the density.

### XPB

The structure of the XPB NTD was also built de novo, the structure of the DRD was extensively rebuilt, and the RecA-like domains were rebuilt to improve the fit to the density.

### XPD

The improved cryo-EM map enabled detailed re-building of XPD, including correction of register shifts in the more poorly ordered regions of the protein and extension of the N- and C-termini.

## MAT1

The N-terminal MAT1 RING domain (*Gervais et al., 2001*) was first docked into the focused classified density and combined with the rest of the TFIIH model, were it helped guide the assignment of the sequence register to the MAT1 model, in combination with density features of large side chains in the helical regions.

## Building of p34

The human p34 structure (*Radu et al., 2017*) was docked into the map as is, extended near the interaction site with p52, and combined with a completely re-built model of the C-terminal eZnF domain (*Schilbach et al., 2017*).

## Building of p44

The p44 VWFA fold needed only minor rebuilding and was combined with the eZnF domain and the C-terminal human RING domain model (*Radu et al., 2017*; *Schilbach et al., 2017*). The p44 NTE was built according to the density at the contact site with XPB and guided by CX-MS data (*Luo et al., 2015*). Both the features of the cryo-EM map and crosslinks of the p44 NTE to p34, p52, and XPB (*Figure 1—figure supplement 6E*) unambiguously confirm the tracing of this segment towards the XPB NTD, rather than alternative tracing towards XPD (this density is now assigned to p62, in agreement with p62-XPD crosslinks; see below and *Figure 1—figure supplement 6G*).

## Building of p62

The p62 protein was modeled based on the placement of the BSD domains (PDB ID 2DII), secondary structure prediction, extension of docked coordinates (*Greber et al., 2017*; *Schilbach et al., 2017*), and new tracing of the protein chain. Placement of the regions near XPD, where density is weak overall, was guided by matching the succession of secondary structure elements along the p62 sequence with helical densities in the cryo-EM map (*Figure 2—figure supplement 1A*), and corroborated by CX-MS data (*Luo et al., 2015*), which showed excellent agreement of p62-XPD crosslinking data with the structure (*Figure 1—figure supplement 6G*). Crosslinks between p62 and p44 showed a relatively large proportion of outliers (*Supplementary file 4*), which may be due to the fact that the sequence register of the p62 segments to which these crosslinks map is not well constrained. These segments were modeled as poly-alanines and deposited without sequence assignment (UNK; *Supplementary file 2*). Maps low-pass filtered to 6 Å and sharpened by a B-factor of only $-100$ Å$^2$ were used to guide docking of domains and assess the continuity of the density in poorly ordered regions of the protein (*Figure 2—figure supplement 1B,C*).

The resulting coordinate model (*Supplementary file 2*) was refined against the final overall reconstruction at 3.7 Å resolution using the real space refinement program in PHENIX (*Adams et al., 2010*; *Afonine et al., 2018*) and validated using the MTRIAGE program in PHENIX and the MOL-PROBITY web server (*Chen et al., 2010*). Ramachandran, C$_\beta$, rotamer, and secondary structure restraints were used throughout the refinement to ensure good model geometry at the given resolution. Data used in the refinement excluded spatial frequencies beyond the nominal 3.7 Å resolution of the cryo-EM map to prevent over-fitting. Additionally, by specifically monitoring the bond length and bond angle r.m.s.d. values, the real space refinement program in PHENIX automatically estimates the relative weighing of the restraint and map data to maintain good model geometry and to prevent over-refinement of the structure (*Adams et al., 2010*; *Afonine et al., 2018*). Because the automatically determined weight fluctuated between approximately 3 and 6 during a typical refinement run, we used the average value of 4.5 for the final refinement (five macro cycles of global optimization and B-factor refinement). The N-terminal RING domain of MAT1 and the BSD1 domain of p62, for which only poorly resolved density is present in the final cryo-EM map, were additionally restrained by reference restraints (*Headd et al., 2012*) using the NMR structures of the corresponding domains (PDB ID 1G25 and 2DII, respectively) (*Gervais et al., 2001*). The side chains of these two domains (with the exception of residues involved in zinc finger formation and of prolines) were truncated at the C$\beta$ position to reflect the lower resolution of the corresponding densities. The FSC curve between the refined coordinate model and the cryo-EM map extends to 3.9 Å and the distribution of B-factors in the refined coordinate model (*Figure 1—figure supplement 2C*) mirror the local resolution of the cryo-EM map (*Figure 1—figure supplement 2D*), as expected. Refinement

statistics are given in *Supplementary file 3* and are typical for structures in this resolution range (100[th] percentile for MOLPROBITY clash score and overall score (*Chen et al., 2010*)).

## Flexibility analysis

For the analysis of conformational dynamics of TFIIH, VPP datasets 1 and 2 were subjected to multi-body refinement in RELION 3 (*Nakane et al., 2018*) using six masks (*Figure 4—figure supplement 1*). After completion of multi-body refinement, we used RELION three to run a principal component analysis to identify the principal modes of motion of the bodies relative to each other (*Nakane et al., 2018*). The volume series for the first 12 principal components were reconstructed and difference densities (green and purple in *Figure 4—figure supplement 1*) were computed between the most extreme states in each series and are shown in *Figure 4—figure supplement 1*. Subsequently, roughly 20,000 particles corresponding to both ends of the distribution were used for selected principal components and subjected to 3D refinement, resulting in maps of approx. 10 Å resolution. It is important to note that the particles used for these refinements, and the subsequent analysis shown in *Figure 4A*, were un-subtracted original particle images containing the entire TFIIH. These refinements are therefore not directly affected by any limitations on alignment accuracy that would arise from alignment of smaller sub-volumes of TFIIH. We also repeated this analysis for two different data subsets (the final 138,659 particle-subset that gave rise to the 3.7 Å-resolution reconstruction and the complete set of 786,755 VPP particles resulting from the initial 3D classification) using only three bodies for multibody refinement (providing more signal for alignment per body) and obtained consistent results overall, with the exception that the ranking of the principal components changed in some instances.

The refined atomic model of the TFIIH core complex, subdivided into suitable rigid bodies, was then rigid-body refined into these volumes using PHENIX real space refinement (*Afonine et al., 2018*) and coordinate displacement between the two resulting models for each principal component was plotted to obtain an initial assessment of the modes of motion present in the TFIIH dataset (*Figure 4A*). For actual structural interpretation (*Figure 4B,C*), the final cryo-EM maps of the TFIIH core complex (this work) and TFIIH in the context of the Pol II-PIC (*Schilbach et al., 2017*) were used.

## Other

Figures were created using PyMol (The PyMOL Molecular Graphics System, Version 1.8 Schrödinger, LLC.) and the UCSF Chimera package from the Computer Graphics Laboratory, University of California, San Francisco (supported by NIH P41 RR-01081) (*Pettersen et al., 2004*). Protein-protein interface statistics were determined using PISA (*Krissinel and Henrick, 2007*). Multiple sequence alignments were performed with Clustal Omega (*Sievers et al., 2011*).

## Data availability

The cryo-EM map of the human TFIIH core complex at 3.7 Å and the refined coordinate model have been deposited to the EMDB and PDB with accession codes EMD-0452 and PDB-6NMI, respectively. Additional cryo-EM maps resulting from the classification of the dataset for presence of the MAT1 RING domain and for the p62 BSD2 domain (both presence and absence) have been deposited to the EMDB with accession codes EMD-0587, EMD-0589, and EMD-0588, respectively. The multibody-refined maps for XPD-MAT1, XPB-p8-p52 (clutch, CTD), and p44-p34-p62-p52 (N-terminal region) have been deposited with accession codes EMD-0602, EMD-0603, and EMD-0604, respectively.

## Acknowledgments

We thank D Wigley for sharing of structural data, S Scheres for sharing computer code before publication, R Tjian, S Zheng, and D King for supplying XPB mAB and peptides for TFIIH purification, P Grob for microscopy support, A Chintangal and P Tobias for computing support, and Z Yu and C Hong for data collection at the Janelia Research Campus cryo-EM facility. We acknowledge the use of the resources of the National Energy Research Scientific Computing Center (NERSC), a DOE Office of Science user facility supported by the Office of Science of the U.S. Department of Energy under Contract No. DE-AC02-05CH11231. This work was funded through NIGMS grants R01-

GM63072, P01-GM063210, and R35-GM127018 to EN. BJG was supported by fellowships from the Swiss National Science Foundation (projects P300PA_160983, P300PA_174355). EN is a Howard Hughes medical investigator.

## Additional information

### Funding

| Funder | Grant reference number | Author |
| --- | --- | --- |
| National Institute of General Medical Sciences | R01-GM63072 | Eva Nogales |
| Howard Hughes Medical Institute | | Eva Nogales |
| Swiss National Science Foundation | Advanced PostDoc Mobility Fellowship P300PA_160983 | Basil J Greber |
| National Institute of General Medical Sciences | R35-GM127018 | Eva Nogales |
| National Institute of General Medical Sciences | P01-GM063210 | Eva Nogales |
| Swiss National Science Foundation | Advanced PostDoc Mobility Fellowship P300PA_174355 | Basil J Greber |

The funders had no role in study design, data collection and interpretation, or the decision to submit the work for publication.

### Author contributions

Basil J Greber, Investigation, Writing—original draft, Performed cryo-EM specimen preparation, data processing, atomic model building, and coordinate refinement, Contributed to data collection; Daniel B Toso, Investigation, Contributed to data collection; Jie Fang, Resources, Investigation, Performed HeLa cell culture and purified TFIIH; Eva Nogales, Supervision, Funding acquisition, Writing—review and editing, Directed the study

### Author ORCIDs

Basil J Greber http://orcid.org/0000-0001-9379-7159
Eva Nogales http://orcid.org/0000-0001-9816-3681

### Decision letter and Author response

Decision letter https://doi.org/10.7554/eLife.44771.051
Author response https://doi.org/10.7554/eLife.44771.052

## Additional files

### Supplementary files

• Supplementary file 1. Data collection statistics. All datasets were acquired on Gatan K2 Summit direct electron detectors mounted in 300 kV-electron microscopes with three-condenser type electron optics. The high rejection rate for the data collected without VPP was due to poorer CTF resolution estimates compared to the VPP data, likely because of the use of a low-base Titan with a less stable side entry holder for most of these data. Abbreviations: TEM, transmission electron microscope; VPP, volta phase plate; $\Sigma$, sum.
DOI: https://doi.org/10.7554/eLife.44771.027

• Supplementary file 2. Components of the TFIIH structure.
DOI: https://doi.org/10.7554/eLife.44771.028

• Supplementary file 3. Data collection, map and model refinement, model validation.
DOI: https://doi.org/10.7554/eLife.44771.029

• Supplementary file 4. Result summary of mapping of CX-MS data (*Luo et al., 2015*) onto the TFIIH structure.
DOI: https://doi.org/10.7554/eLife.44771.030

• Transparent reporting form
DOI: https://doi.org/10.7554/eLife.44771.032

## Data availability

The cryo-EM map of the human TFIIH core complex at 3.7 Å and the refined coordinate model have been deposited to the EMDB and PDB with accession codes EMD-0452 and PDB-6NMI, respectively. Additional cryo-EM maps resulting from the classification of the dataset for presence of the MAT1 RING domain and for the p62 BSD2 domain (both presence and absence) have been deposited to the EMDB with accession codes EMD-0587, EMD-0589, and EMD-0588, respectively. The multibody-refined maps for XPD-MAT1, XPB-p8-p52 (clutch, CTD), and p44-p34-p62-p52 (N-terminal region) have been deposited with accession codes EMD-0602, EMD-0603, and EMD-0604, respectively.

The following datasets were generated:

| Author(s) | Year | Dataset title | Dataset URL | Database and Identifier |
|---|---|---|---|---|
| Greber BJ, Toso DB, Fang J, Nogales E | 2019 | Cryo-EM map of the human TFIIH core complex at 3.7 Å | http://www.ebi.ac.uk/pdbe/entry/emdb/EMD-0452 | Electron Microscopy Data Bank, EMD-0452 |
| Greber BJ, Toso DB, Fang J, Nogales E | 2019 | Refined coordinate model of the human TFIIH core complex at 3.7 Å | http://www.rcsb.org/structure/6NMI | Protein Data Bank, 6NMI |
| Greber BJ, Toso DB, Fang J, Nogales E | 2019 | Cryo-EM map resulting from the classification of the dataset for presence of the MAT1 RING domain | http://www.ebi.ac.uk/pdbe/entry/emdb/EMD-0587 | Electron Microscopy Data Bank, EMD-0 587 |
| Greber BJ, Toso DB, Fang J, Nogales E | 2019 | Cryo-EM map resulting from the classification of the dataset for presence of the p62 BSD2 domain | http://www.ebi.ac.uk/pdbe/entry/emdb/EMD-0589 | Electron Microscopy Data Bank, EMD-0 589 |
| Greber BJ, Toso DB, Fang J, Nogales E | 2019 | Cryo-EM map resulting from the classification of the dataset for absence of the p62 BSD2 domain | http://www.ebi.ac.uk/pdbe/entry/emdb/EMD-0588 | Electron Microscopy Data Bank, EMD-0 588 |
| Greber BJ, Toso DB, Fang J, Nogales E | 2019 | Multibody-refined map for XPD-MAT1 | http://www.ebi.ac.uk/pdbe/entry/emdb/EMD-0602 | Electron Microscopy Data Bank, EMD-060 2 |
| Greber BJ, Toso DB, Fang J, Nogales E | 2019 | Multibody-refined map for XPB-p8-p52 (clutch, CTD) | http://www.ebi.ac.uk/pdbe/entry/emdb/EMD-0603 | Electron Microscopy Data Bank, EMD-060 3 |
| Greber BJ | 2019 | Multibody-refined map for p44-p34-p62-p52 (N-terminal region) | http://www.ebi.ac.uk/pdbe/entry/emdb/EMD-0604 | Electron Microscopy Data Bank, EMD-0604 |

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

## Appendix 1

DOI: https://doi.org/10.7554/eLife.44771.033

# The XPD-p44 interface

XPD serves a dual role within TFIIH: Its helicase activity is essential for TFIIH function in NER (*Kuper et al., 2014*; *Winkler et al., 2000*), and it serves as an architectural element in transcription, enabling the proper placement of the CAK subcomplex within the Pol II PIC (*Abdulrahman et al., 2013*; *Drapkin et al., 1996*; *Dubaele et al., 2003*; *Greber et al., 2017*; *Guzder et al., 1994*; *He et al., 2016*; *Murakami et al., 2015*; *Robinson et al., 2016*; *Rossignol et al., 1997*; *Schilbach et al., 2017*; *Tirode et al., 1999*; *Tsai et al., 2017*). The XPD-p44 interaction is critical both for the enzymatic and architectural roles of XPD (*Coin et al., 1998*; *Dubaele et al., 2003*; *Kim et al., 2015*; *Kuper et al., 2014*). Therefore, disruption of the XPD-p44 interface can simultaneously cause NER defects and impair transcription initiation, which leads to a TTD phenotype in human patients (*Dubaele et al., 2003*). Our structure provides a detailed view of this interaction interface, which encompasses the p44 loops β1-α1, β4-α5, β5-α6, and β6-α7 connecting β-strands and α-helices in the p44 VWA domain (*Figure 5E*, *Figure 5—figure supplement 2*). Mutations L174W, T175R, and D178A in the p44 β4-α5 loop have been shown to abrogate the p44-XPD interaction and to reduce the helicase activity of XPD, implicating this region in XPD activation (*Kim et al., 2015*; *Seroz et al., 2000*). Our structure suggests that the insertion of large residues (L174W, T175R) into the tightly packed p44-XPD interface (*Figure 5—figure supplement 2B*) is likely to lead to steric clashes, while the D178A mutation would break a likely salt bridge with XPD R592 (*Figure 5—figure supplement 2B*). Strikingly, the R592P mutation in XPD, the side chain of which packs against the p44 β4-α5 loop in wild type XPD, causes TTD in human patients (*Cleaver et al., 1999*), underscoring the importance of this interaction.

In XPD, RecA2 loop residues 531–533, 563–565, 589–592, 612–616, and the helical segment 718–725 form the p44 interaction interface (*Figure 5E*). Using recombinant human and *Chaetomium thermophilum* proteins, it has been shown that the XPD R722W mutation impairs the interaction between p44 and XPD, thereby interfering with both the p44-dependent stimulation of the XPD ATPase and helicase activities in NER, and architectural role of p44 in transcription initiation (*Kuper et al., 2014*). Our structure shows that the R722W mutation disrupts a salt bridge to p44 D75, and the bulky tryptophan side chain may additionally cause steric clashes with neighboring p44 residues, explaining the deleterious effect of the mutation (*Figure 5E*, *Figure 5—figure supplement 2C*). It has been proposed that the R616P, D673G, and G675R disease mutations, which impair the p44-XPD interaction and the XPD helicase activity (*Dubaele et al., 2003*), exert their deleterious effect by impairing the interaction between XPD and p62 (*Luo et al., 2015*). However, our structure shows that G675 is buried in the core of XPD (*Figure 5—figure supplement 2D*), where accommodation of the large arginine side chain in G675R will disrupt the structure of the protein. XPD D673 is located in a β-strand in the vicinity of the XPD-p62 interface (*Figure 5—figure supplement 2D*); however, no direct interactions of this residue with p62 are observed. XPD R616 is located at the p44-XPD interface, where its side chain is packed against neighboring residues (e.g. Y674, located between the D673 and G675, the sites of the D673G and G675R mutations). R616 or Y674 may engage in interactions with p44 E204 (*Figure 5E* and *Figure 5—figure supplement 2D*). These interactions would be disrupted in the R616P mutant. Altogether, these observations suggest that these mutations act either via disruption of the XPD structure or the XPD-p44 interface, not via p62. Nevertheless, a contribution of p62 to the stabilization of the p44-XPD interface cannot be excluded because several p62 elements are observed near the interface (*Figure 5—figure supplement 2E*). As discussed in the main text, some of these interactions are dynamic (*Figure 4—figure supplement 2G–J*) and unlikely to be strictly required for the stability of the p44-XPD interface.

## Appendix 2

DOI: https://doi.org/10.7554/eLife.44771.033

### Mapping of XPD mutations

In addition to our analysis of the XPD-p44 interface and the mutations located in its vicinity, we were also able to map numerous other human disease mutations onto our structure (*Figure 6*). Our results are in good agreement with the hypotheses derived from comparative modeling based on homologous bacterial DNA repair enzymes (*Bienstock et al., 2003*), archaeal XPD homologs (*Fan et al., 2008*; *Liu et al., 2008*; *Wolski et al., 2008*), and our previous lower-resolution TFIIH structure (*Greber et al., 2017*). Several XP mutations map in the regions that interact with bound DNA (*Figure 6—figure supplement 1A*), as confirmed by superposing homologous DNA-bound helicase structures (*Cheng and Wigley, 2018*), or in the conserved motifs near the ATPase active site (*Figure 6—figure supplement 1B*). These mutations impair XPD helicase activity (*Dubaele et al., 2003*; *Fan et al., 2008*; *Liu et al., 2008*), causing the NER defects that manifest in the XP disease symptoms. As explained above, TTD mutations tend to map to the periphery of XPD, including the XPD-p44 interface (*Figure 5E*, *Figure 5—figure supplement 2*, *Figure 6—figure supplement 1C,D*), the ARCH domain (*Figure 6—figure supplement 1E*), or the FeS domain (*Figures 5B* and *6*). The C259Y mutation in the ARCH domain likely impairs the proper folding of this domain due to steric hindrance introduced by the large tyrosine side chain, which in turn impairs association of the CAK subcomplex through disruption of MAT1 association, as discussed previously (*Greber et al., 2017*; *Liu et al., 2008*; *Wolski et al., 2008*). Consistent with its observed location in our structure (*Figure 5B*), the R112H mutation in the FeS domain leads to destabilization of the 4FeS cluster (*Rudolf et al., 2006*), which in turn leads to the disruption of folding of this entire domain (*Fan et al., 2008*) and lower TFIIH levels overall (*Botta et al., 2002*). Another TTD mutation that does not localize to the periphery of XPD is the particularly interesting temperature-sensitive mutation R658C, for which patients show exacerbated symptoms, including complete hair loss, when they develop a fever (*Vermeulen et al., 2001*). This residue is located at the interface between the XPD RecA1 and RecA2 domains, where it forms a salt bridge across the domain interface (*Figure 6B*). This interface would be partially destabilized by the mutation at normal temperature and more severely disrupted at elevated temperature, leading to stronger disease symptoms.

We also mapped onto our structure the location of the so-called *rem* mutations in Rad3, the yeast homolog of XPD. These mutations increase the affinity of Rad3 for DNA, which leads to retention of Rad3/XPD on DNA lesions, incomplete NER reactions, and subsequent DNA breakage upon arrival of a DNA replication fork (*Herrera-Moyano et al., 2014*; *Moriel-Carretero and Aguilera, 2010*). We mapped the *rem* mutations Rad3-E236G (XPD E235), Rad3-A237T (XPD A236), and Rad3-H661Y (XPD H659), as well as the human XP-CS mutation XPD-G675R, which also exhibits elevated DNA affinity but reduced helicase activity in an archaeal model system (*Fan et al., 2008*) and DNA breakage in patient cells (*Theron et al., 2005*), onto our structure (*Figure 6—figure supplement 1F*). XPD E235 and A236 are part of the conserved helicase motif II (signature sequence DEAH, *Figure 5—figure supplement 1C*, *Figure 6—figure supplement 1F*) (*Fairman-Williams et al., 2010*), and as such critical for nucleotide hydrolysis. Notably, the neighboring XPD D234, also part of motif II (*Figure 5—figure supplement 1C*, *Figure 6—figure supplement 1B,F*), is a human XP mutation (*Cleaver et al., 1999*). XPD H659 is located nearby, and contacts R601, which locates to the immediate vicinity of the DNA based on superposition with structures of DNA-bound helicases (*Büttner et al., 2007*; *Cheng and Wigley, 2018*) (*Figure 6—figure supplement 1F*) and is affected by an XP mutation (*Cleaver et al., 1999*). In agreement with previous proposals, the *rem* mutations are therefore localized near regions responsible for ATP hydrolysis and DNA binding, consistent with the idea that they lead to stalling of NER complexes at DNA lesions, which may ultimately lead to incomplete NER, DNA breaks, failure to re-start transcription, and the XP-CS phenotype (*Moriel-Carretero et al., 2015*).

In summary, our structure pinpoints the locations of XPD mutations that provide insight into human disease mechanisms, and in most cases resolves the density for the side chains of the affected residues, thereby providing a structural framework for the detailed analysis of the interactions of the affected residues and the effects of these mutations.

