## [Decision Letter]

Thank you for submitting your article "The complete structure of the human TFIIH core complex" for consideration by *eLife*. Your article has been reviewed by three peer reviewers, including Nikolaus Grigorieff as the Reviewing Editor and Reviewer #1, and the evaluation has been overseen by Cynthia Wolberger as the Senior Editor. The following individuals involved in review of your submission have also agreed to reveal their identity: Seth A Darst (Reviewer #2); James M Berger (Reviewer #3).

The reviewers have discussed the reviews with one another and the Reviewing Editor has drafted this decision to help you prepare a revised submission.

Summary:

This is a comprehensive study of the core structure of TFIIH, a transcription factor that acts also as a DNA repair enzyme. The authors present an atomic model, built into a cryo-EM map, that contains eight out of the ten core complex subunits, some incomplete due to partial disorder in the subunits. This new structure explains numerous biochemical and functional data found in the literature, including details of TFIIH assembly and regulation of the DNA repair activity. The authors resolve several critical interactions that knit the complex together and they are able to map human disease mutations onto the assembly to rationalize their molecular effects. This study will be a key contribution to both the eukaryotic transcription and DNA repair fields.

Minor points:

1) What is meant by "extreme" motion?

2) Subsection “Structure determination of TFIIH”: Please provide the full range of resolutions, not just the best value.

3) What is meant by "initial 3D classification"?

4) Data collection: Please include a figure showing a typical micrograph and a set of 2D class averages.

5) Data analysis: The authors mention a problem of preferred orientation. Please show a figure plotting orientations.

6) Figure 4 and Figure 4—figure supplement 1: The visualization of the flexibility results is difficult to follow. In Figure 4, would it be possible to magnify the structures in panel A and perhaps eliminate the light and dark blue dots in areas that do not move much? In the supplement, it might be better to limit the display of the densities to two colors, or maybe consider showing 2D sections instead to highlight the flexible domains. Enlarging the figure may also help.

7) Overall, it would be helpful to have videos that clarify numerous aspects of the structural findings, such as subunit organization (Figure 1), domain motions (Figure 4), and the positions of various disease-linked mutations.

8) Figure 1D. Please indicate what the lines mean between subunits; presumably crosslinks?

9) Close-ups: In a number of figures, close-ups are shown next to overviews of a subunit or the complex (for example, Figure 5A and B, Figure 6A and B, but also elsewhere). It would help if there is some indication in the overviews of where the close-ups are located.

10) Multi-body refinement: The authors state that this did not improve the resolution of the map much and led to only a modest (albeit important) improvement of map features, and that this might indicate increased resolution-limiting alignment errors of the domains included in each body. Since the flexibility analysis presented in Figure 4 and Figure 4—figure supplement 1 relies on these alignment, how reliable is this analysis? Looking at the domain movements indicated in Figure 4, it looks as if some of them represent rotations around the domain centers. Could this be merely an artifact of data processing from rotational blurring as implemented by the maximum likelihood approach? The authors should discuss this and, if appropriate, reevaluate these results.

11) Atomic model refinement: Which map was the model refined against? Was it one of the half maps or the full map? The authors should discuss how they limited over-refinement.

12) About 50% of the videos were discarded. This number seems high. The authors indicate that some images did not show CTF fringes to high resolution, and that their data selection was quite stringent. However, a 3.6-Å reconstruction does not impose much of a limit on the data quality. What was the reason for so many bad images, and what were the quantitative criteria for inclusion/rejection?

13) Of the 2 million particles picked initially, only 7% ended up in the final reconstruction. How do the authors know that these are the "correct" 7%? Could they have missed a major conformation that would have told a very different story?

---

## [Author Response]

Summary:This is a comprehensive study of the core structure of TFIIH, a transcription factor that acts also as a DNA repair enzyme. The authors present an atomic model, built into a cryo-EM map, that contains eight out of the ten core complex subunits, some incomplete due to partial disorder in the subunits. This new structure explains numerous biochemical and functional data found in the literature, including details of TFIIH assembly and regulation of the DNA repair activity. The authors resolve several critical interactions that knit the complex together and they are able to map human disease mutations onto the assembly to rationalize their molecular effects. This study will be a key contribution to both the eukaryotic transcription and DNA repair fields.

We thank the reviewers and editors for the positive assessment of our manuscript. We will address the detailed comments in point-by-point answers below. Other than the changes required for addressing these questions and comments, we have made the following changes and improvements to the manuscript:

i) We previously deposited the main cryo-EM map and coordinate model at the EMDB and PDB. We have now deposited six additional cryo-EM maps to the EMDB: The maps classified for the p62 BSD2 domain (EMD-0588, EMD-0589), the density classified for the *MAT1* RING domain (EMD-0587), and the multibody-refined best-quality maps (EMD-0602, EMD-0603, EMD-0604). The accession codes are provided in the appropriate location in the manuscript (section “Data availability” at the end of the manuscript).

ii) Supplementary files 2 and 3 were updated with the accession codes of the map and model deposited at the EMDB and PDB as well as with the final refinement statistics of the model (Supplementary file 3) and the list of residues deposited as UNK (Supplementary file 2). The statistics are slightly improved overall, e.g. the Ramachandran outliers are down to 0.2% from 0.3% and the bond angle RMSD is down to 1.08° from 1.19°.

iii) Figure 1—figure supplement 2: Panel B was updated with a model vs. map FSC curve computed against a cryo-EM map filtered at 3.5 Å resolution. The lack of a hard cut-off at 3.7 Å resolution results in a more natural decay of the model vs. map curve at high resolutions (because the coordinates correlate with the map beyond the nominal map resolution in the core of the structure), while the resolution estimates are unchanged (both of these observations are as expected). Panel C was updated with the B-factors of the final coordinate model deposited at the PDB.

iv) Figure 1—figure supplement 5: While the manuscript was under review, it was brought to our attention that post-processing unfiltered half maps from intermediate steps of multibody refinement runs can lead to artifacts in the FSC curves at very high resolution (close to Nyquist frequency) and is discouraged. Even though to the best of our knowledge, these issues did not affect our reconstructions in any way, we opted to completely avoid any risk of artifacts. We therefore replaced the initially used multibody-refined maps with maps resulting from a regular final refinement cycle in RELION 3 (using data out to Nyquist frequency), which led to the same improvements in map interpretability observed before, but slightly different resolution estimates (3.6-3.8 Å depending on the sub-volume; see also the point-to-point answer on that topic below). These have been updated in the figure (along with the FSC curves).

v) Figure 4C and D now use more accurate representations of the “open” conformation of TFIIH, based on fitting of component domains of our human TFIIH structure into the cryo-EM maps of the human and yeast Pol II-PIC. The legend has been adjusted accordingly; the information content of the figures is otherwise unchanged and no changes to the text were needed.

Minor points:1) What is meant by "extreme" motion?

We reworded this:

“… because it is flexibly tethered to the TFIIH core complex”.

2) Subsection “Structure determination of TFIIH”: Please provide the full range of resolutions, not just the best value.

While the manuscript was under review, we modified the way the multi-body refinement was conducted (see general comments above). The XPD-*MAT1* body is now resolved at 3.6 Å resolution, while the remaining two bodies did not see an improvement in resolution (even though the map quality increased in certain regions in all bodies). This has been clarified in the manuscript:

“…, which resulted in density maps of improved interpretability for all three sub-volumes and a slightly improved resolution of 3.6 Å for the XPD-MAT1 region.”

3) What is meant by "initial 3D classification"?

We refer to the initial 3D classification to remove broken particles and false-positive particle picks, as introduced in the preceding sentence. We have clarified this by specifying the angular sampling used, and mentioning that this classification was a coarse angular sampling, low-resolution classification:

“To remove false positive particle picks and broken particles, an initial 3D classification at low resolution (7.5° angular sampling) was performed on each dataset individually (datasets 3, 4, 5) […]). In summary, a total of > 2,000,000 initial particle picks were subjected to this initial low-resolution 3D classification, identifying approx. 820,000 intact particles that were subjected to further processing.”

4) Data collection: Please include a figure showing a typical micrograph and a set of 2D class averages.

These panels have been added to Figure 1—figure supplement 1A and B.

5) Data analysis: The authors mention a problem of preferred orientation. Please show a figure plotting orientations.

We have added a representation of the orientation distribution to Figure 1—figure supplement 1D. The orientations populate mostly a zone that corresponds to particles adhering to the carbon support with the long axis of the complex parallel to the support layer, and a range of orientations arising from rotation of the particles around that long axis. Dataset 6 showed stronger orientation bias compared to the other datasets. Therefore, we pooled these particles with a different dataset for initial 3D classification to avoid bias in classification at this step. Subsequently, the resulting particles from dataset 6 were joined with the remaining datasets.

The legend to the new figure panel reads:

“(D) Depiction of the orientation distribution of the particle images in the final reconstruction. As also evident from the 2D classes (C), the orientations populate mostly a zone that corresponds to particles adhering to the carbon support with long axis of the complex parallel to the support layer, and a range of orientations arising from rotation of the particles around that long axis.”

6) Figure 4 and Figure 4—figure supplement 1: The visualization of the flexibility results is difficult to follow. In Figure 4, would it be possible to magnify the structures in panel A and perhaps eliminate the light and dark blue dots in areas that do not move much?

We have modified this figure according to the suggestion. We have removed all of the vectors shorter than 2.5 Å and rearranged the panels such that the figure can appear larger in the typeset version of the article.

In the supplement, it might be better to limit the display of the densities to two colors, or maybe consider showing 2D sections instead to highlight the flexible domains. Enlarging the figure may also help.

We have visually simplified the figure by using the same color for the two volumes corresponding to the ends of the eigenvector movie and additionally, we enlarged the panels by removing 4 panels representing eigenvectors that are not used or discussed further in the manuscript.

7) Overall, it would be helpful to have videos that clarify numerous aspects of the structural findings, such as subunit organization (Figure 1), domain motions (Figure 4), and the positions of various disease-linked mutations.

We have added three videos on the overall architecture of the complex (Video 1), conformational changes during incorporation into the transcription pre-initiation complex (Video 2), and mapping of disease mutations (Video 3). We hope that these videos will help both reviewers and readers visualize the spatial arrangement of the structural elements discussed in the paper.

8) Figure 1D. Please indicate what the lines mean between subunits; presumably crosslinks?

The lines do not indicate crosslinks, but actual protein-protein interactions derived from the structure, mapped at the domain level. Elucidating which domains of which TFIIH subunits interact with each other has been the goal of a number of previous publications in the field, and this figure panel provides the answer to this question. We have emphasized this point in the figure legend:

"Domain-level protein-protein interaction network between the components of the TFIIH core complex and MAT1 derived from the interactions observed in our structure."9) Close-ups: In a number of figures, close-ups are shown next to overviews of a subunit or the complex (for example, Figure 5A and B, Figure 6A and B, but also elsewhere). It would help if there is some indication in the overviews of where the close-ups are located.

We have added this information to the figures mentioned (Figure 5A, 6A).

10) Multi-body refinement: The authors state that this did not improve the resolution of the map much and led to only a modest (albeit important) improvement of map features, and that this might indicate increased resolution-limiting alignment errors of the domains included in each body.

Decreasing alignment power with decreasing molecular weight (and decreasing total available signal) of the complex under study is a problem that is commonly encountered in single particle cryo-EM structure determination, and is one of the reasons for why phase plates such as the VPP have been developed (e.g. Khoshouei et al., 2017, as cited in the manuscript).

Since the flexibility analysis presented in Figure 4 and Figure 4—figure supplement 1 relies on these alignment, how reliable is this analysis? Looking at the domain movements indicated in Figure 4, it looks as if some of them represent rotations around the domain centers. Could this be merely an artifact of data processing from rotational blurring as implemented by the maximum likelihood approach? The authors should discuss this and, if appropriate, reevaluate these results.

The reviewers bring up an important point. However, we believe that our approach should eliminate the risk of generating this type of artifact. It is important to note that after multi-body refinement (alignment of sub-volumes) and principal component analysis, we selected the complete, non-subtracted particle images (showing entire TFIIH complexes) corresponding to the tails of the distribution (i.e. largest conformational difference) and re-refined these small datasets. Coordinates were then rigid body refined into these volumes, and the positional differences of these fitted coordinates is what is plotted in Figure 4A. Therefore, importantly, the analysis and conclusions shown in Figure 4 do not directly rely on the alignment of the sub-volumes. The alignment of sub-volumes only gave the principal components according to which the particle images for re-refinement were selected. If this selection had been based merely on alignment errors of the sub-volumes without any actual conformational differences occurring in the actual particle images, we would not expect to find these same conformational differences in the volumes obtained from refinement of complete particle images. We clarified and elaborated on this point in the Materials and methods section:

"It is important to note that the particles used for these refinements, and the subsequent analysis shown in Figure 4A, were un-subtracted original particle images containing the entire TFIIH. […] We also repeated this analysis for two different data subsets (the final 138,659 particle-subset that gave rise to the 3.7 Å-resolution reconstruction and the complete set of 786,755 VPP particles resulting from the initial 3D classification) using only 3 bodies for multibody refinement (providing more signal for alignment per body) and obtained consistent results overall, with the exception that the ranking of the principal components changed in some instances."

11) Atomic model refinement: Which map was the model refined against? Was it one of the half maps or the full map? The authors should discuss how they limited over-refinement.

The model was refined against the full map to utilize the full signal available in the cryo-EM dataset. We estimate that we would lose approximately 0.3-0.5 Å resolution if we were to use only a half-map for refinement.

The real space refinement algorithm in PHENIX automatically estimates the relative weighing of geometry restraints and density (Adams et al., 2010; Afonine et al., 2018; as cited in the manuscript) based on the bond length and bond angle RMSD values to prevent overfitting. Because these automated target weights fluctuated between refinement cycles, sometimes giving rise to arbitrarily higher or lower weights in the last iteration (which determines the final fit to map and geometry quality), we used a weight of 4.5 for the final 5 macro-cycles of refinement, corresponding to the average of weights estimated by PHENIX during a trial run on the same data. These settings resulted in bond length and bond angle RMSD values in the expected range (1.08° for bond angles, 0.01 Å for bond lengths). We have added this information in the Materials and methods section of the manuscript:

"Additionally, by specifically monitoring the bond length and bond angle r.m.s.d values, the real space refinement program in PHENIX automatically estimates the relative weighing of the restraint and map data to maintain good model geometry and to prevent over-refinement of the structure (Adams et al., 2010; Afonine et al., 2018). Because the automatically determined weight fluctuated between approximately 3 and 6 during a typical refinement run, we used the average value of 4.5 for the final refinement (5 macro cycles of global optimization and B-factor refinement)."

As detailed in the manuscript, we additionally used reference restraints, where available, to avoid overfitting of coordinates in poorly resolved regions of the cryo-EM map.

12) About 50% of the videos were discarded. This number seems high. The authors indicate that some images did not show CTF fringes to high resolution, and that their data selection was quite stringent. However, a 3.6-Å reconstruction does not impose much of a limit on the data quality. What was the reason for so many bad images, and what were the quantitative criteria for inclusion/rejection?

As detailed in Supplementary file 1, the high rejection rate affected mostly the non-VPP data, which was affected by higher variability than the VPP data. Most of the non-VPP data was collected on a low-base FEI Titan (non-KRIOS) microscope with a side-entry holder, where stage drift is a concern, and consequently, more images were rejected due to poor resolution of the CTF fitting (worse than 4.5 Å). Unexpectedly, the data from the Janelia Research Campus Titan KRIOS was affected by similar problems, with large numbers of micrographs exhibiting CTF fit resolutions of only 5 Å. Inclusion of these lower-quality micrographs and combining them with the VPP data degraded the resolution of the reconstructions at an early stage, so stringent selection criteria were applied to the non-VPP data and inclusion of the resulting data improved the reconstruction until the VPP data alone exceeded 3.9 Å resolution, at which point even the best non-VPP data no longer improved the resolution and was consequently removed from the calculations (as described in the manuscript).

We have added a short note to the legend of Supplementary file 1:

"The high rejection rate for the data collected without VPP was due to poorer CTF resolution estimates compared to the VPP data, likely because of the use of a low-base Titan with a less stable side entry holder for most of these data."

13) Of the 2 million particles picked initially, only 7% ended up in the final reconstruction. How do the authors know that these are the "correct" 7%? Could they have missed a major conformation that would have told a very different story?

We are confident that we did not miss a major conformation of TFIIH that would have led to a vastly different interpretation. The 2 million initial particle picks included false positives (e.g. non-particles showing only carbon, or ice contamination) as well as presumably damaged complexes that never resulted in a reconstruction with discernable features. Notably, almost 40% of the initial particles (approx. 800,000 particle images) were retained after the first round of 3D classification and their refinement resulted in a reconstruction that is highly similar to the final reconstruction, albeit at lower resolution (approx. 4.3 Å) and considerably worse map quality because of TFIIH flexibility (mostly motion of XPB that results in opening and closing of the TFIIH “horseshoe”). We have clarified this in the Materials and methods section:

“We note that even though the final reconstruction comprises only a relatively small fraction of the total particle picks, the first 3D refinement from 786,755 VPP particle images (Figure 1—figure supplement 1) resulted in a 4.3 Å-resolution map that is in excellent agreement with the final map, except for lower resolution and worse map quality caused by residual heterogeneity that was addressed in the subsequent 3D classification step to yield the final set of 138,659 particle images. Therefore, we conclude that our final reconstruction is representative of the overall particle population in the dataset.”